# Meta-connectomic analysis maps consistent, reproducible, and transcriptionally relevant functional connectome hubs in the human brain

Zhilei Xu [1,2,3], Mingrui Xia [1,2,3], Xindi Wang[1,2,3], Xuhong Liao[4], Tengda Zhao[1,2,3] & Yong He [1,2,3,5✉]

Human brain connectomes include sets of densely connected hub regions. However, the consistency and reproducibility of functional connectome hubs have not been established to date and the genetic signatures underlying robust hubs remain unknown. Here, we conduct a worldwide harmonized meta-connectomic analysis by pooling resting-state functional MRI data of 5212 healthy young adults across 61 independent cohorts. We identify highly consistent and reproducible connectome hubs in heteromodal and unimodal regions both across cohorts and across individuals, with the greatest effects observed in lateral parietal cortex. These hubs show heterogeneous connectivity profiles and are critical for both intra- and inter-network communications. Using post-mortem transcriptome datasets, we show that as compared to non-hubs, connectome hubs have a spatiotemporally distinctive transcriptomic pattern dominated by genes involved in the neuropeptide signaling pathway, neurodevelopmental processes, and metabolic processes. These results highlight the robustness of macroscopic connectome hubs and their potential cellular and molecular underpinnings, which markedly furthers our understanding of how connectome hubs emerge in development, support complex cognition in health, and are involved in disease.

[1] State Key Laboratory of Cognitive Neuroscience and Learning, Beijing Normal University, Beijing, China. [2] Beijing Key Laboratory of Brain Imaging and Connectomics, Beijing Normal University, Beijing, China. [3] IDG/McGovern Institute for Brain Research, Beijing Normal University, Beijing, China. [4] School of Systems Science, Beijing Normal University, Beijing, China. [5] Chinese Institute for Brain Research, Beijing, China. ✉email: yong.he@bnu.edu.cn

Functional connectome mapping studies have identified sets of densely connected regions in large-scale human brain networks, which are known as hubs[1]. Connectome hubs play a crucial role in global brain communication[1,2] and support a broad range of cognitive processing, such as working memory[3] and semantic processing[4]. Growing evidence suggests that these highly connected brain hubs are preferentially targeted by many neuropsychiatric disorders[5–8], which provides critical clues for understanding the biological mechanisms of disorders and establishing biomarkers for disease diagnosis[8,9] and treatment evaluation[10] (refs. [1,2,11,12] for reviews).

Despite such importance, there is considerable inconsistency in anatomical locations of functional connectome hubs among existing studies. For example, components of the default-mode network (DMN) have been frequently reported as connectome hubs, yet the spatial pattern is highly variable across studies. In particular, several studies have shown highly connected hubs in the lateral parietal regions of the DMN[7,8,13,14], whereas others have reported midline structures of the DMN[15–19]. Several works have identified primary sensorimotor and visual regions as connectome hubs[13,14,16–19], yet others did not replicate these findings[7,8,15]. Subcortical regions, such as the thalamus and amygdala, have also been inconsistently reported as hubs[8,15,16,18] and non-hubs[7,13,14,17,19]. Thus, the consistency and reproducibility of functional connectome hubs have been difficult to establish to date, which can be attributed to inadequate sample size and differences in imaging scanner, imaging protocol, data processing, and connectome analysis strategies. Here, we aimed to establish a harmonized meta-analysis model to identify robust functional connectome hubs in healthy young adults by combining multiple cohorts with uniform protocols for data quality assurance, image processing, and connectome analyses.

Once the robust connectome hubs are identified, we will further examine their genetic signatures. It has been well demonstrated that the connectome architecture of the human brain is inheritable, such as functional connectivity of the DMN[20] and the cost-efficiency optimization[21]. Moreover, the functional connectomes can be regulated by genotypic variation both during rest[22] and in cognitive tasks[23], especially involving the DMN[22,23] and frontoparietal network (FPN)[23]. Growing evidence also suggests spatial correspondence between transcriptomic profiles and connectome architectures[24–26] (ref. [27] for review). Thus, we reasoned that the robust macroscopic connectome hubs could be associated with microscopic genetic signatures. Elucidating these genetic signatures will substantially benefit our understanding of how connectome hubs emerge in development, function in complex cognition, and are involved in disease.

To address these issues, we provided, to the best of our knowledge, the first worldwide harmonized meta-connectomic analysis of functional brain hubs by pooling a large-sample resting-state functional MRI (rsfMRI) dataset of 5212 healthy young adults (aged 18–36 years, 2377 males) across 61 independent cohorts. We identified highly consistent and reproducible functional connectome hubs in multiple heteromodal and unimodal regions, with the most robust findings occurring in several lateral parietal regions. These connectome hubs showed unique and heterogeneous connectivity profiles to provide support for both intra- and inter-network communications. To uncover the genetic signatures underlying these connectome hubs, we conducted machine learning approaches to distinguish connectome hubs from non-hubs using transcriptomic data from the Allen Human Brain Atlas (AHBA), explored their developmental evolutions using the BrainSpan Atlas, and assessed their neural relevance by contextualizing them relative to established neuroimaging patterns. We demonstrated that these robust connectome hubs were associated with a spatiotemporal transcriptomic pattern dominated by genes enriched for the neuropeptide signaling pathway, neurodevelopmental processes, and metabolic processes.

## Results

**Identifying consistent connectome hubs using a harmonized meta-analysis model.** Prior to the meta-analysis, we constructed a voxelwise functional connectome matrix for each individual by computing the Pearson's correlation coefficient between preprocessed rsfMRI time series of all pairs of gray matter voxels (47,619 voxels). Then, the functional connectivity strength (FCS) of each voxel was computed as the sum of connection weights between the given voxel and all the other voxels. This resultant FCS map was further normalized with respect to its mean and standard deviation across voxels[7]. For each cohort, we performed a general linear model on these normalized FCS maps to reduce age and sex effects. As a result, we obtained a mean FCS map and its corresponding variance map for each cohort that were used for subsequent meta-analyses.

To identify the most consistent connectome hubs, we conducted a voxelwise random-effects meta-analysis on the mean and variance FCS maps of the 61 cohorts. Such an analysis addressed the across-cohort heterogeneity of functional connectomes, resulting in a robust FCS pattern (Fig. 1a) and its corresponding standard error (SE) map (Fig. 1b). Then, we identified consistent connectome hubs whose FCS values were significantly ($p < 0.001$, cluster size > 200 mm³) higher than the global mean (i.e., zero) using a voxelwise $Z$ value map computed by dividing the FCS map by the SE map. To determine the statistical significances of these observed $Z$ values, a nonparametric permutation test[28] with 10,000 iterations was performed. Finally, we estimated voxelwise effect sizes using Cohen's $d$ metric computed by dividing the $Z$ value map by the square root of the

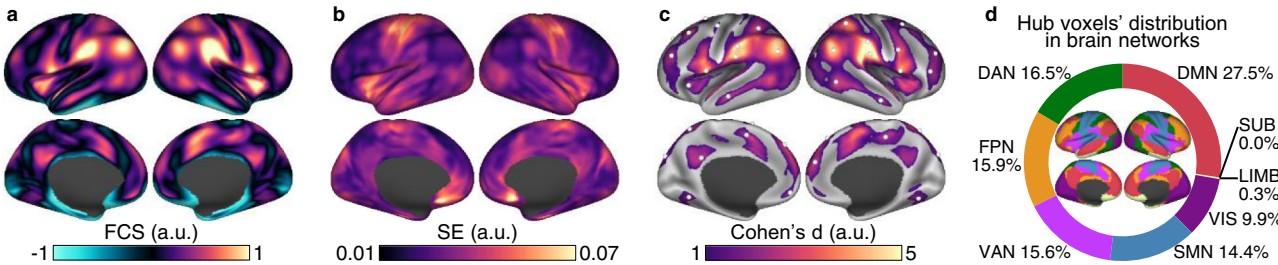

**Fig. 1 Identifying consistent connectome hubs using a harmonized meta-analysis model. a, b** Robust FCS pattern (**a**) and its corresponding variance (standard error, SE) map (**b**) estimated using a harmonized voxelwise random-effects meta-analysis across 61 cohorts. **c** The most consistent functional connectome hubs ($p < 0.001$, cluster size > 200 mm³). White spheres represent hub peaks. **a–c** a.u. arbitrary unit. **d** Hub voxels' distribution in eight large-scale brain networks. Insets depicts the seven large-scale cortical networks[29]. SUB subcortical network, LIMB limbic network.

**Table 1 Highly consistent functional connectome hubs.**

| No. | Hub | Location | MNI coordinates | | | Cohen's d | FCS | SE |
|---|---|---|---|---|---|---|---|---|
| | | | x | y | z | | | |
| 1 | Right PFt | PFt (superoanterior BA 40) | 60 | −21 | 45 | 6.267 | 1.072 | 0.022 |
| 2 | Left PFt | PFt (superoanterior BA 40) | −60 | −24 | 36 | 6.151 | 0.949 | 0.020 |
| 3 | Right PF | PF (posterior BA 40) | 60 | −27 | 24 | 5.785 | 1.239 | 0.027 |
| 4 | Left SCEF | Supplementary and cingulate eye field | 0 | 0 | 51 | 5.635 | 1.000 | 0.023 |
| 5 | Left PGi | PGi (inferior BA 39) | −51 | −66 | 30 | 5.168 | 1.075 | 0.027 |
| 6 | Left PFop | PF opercular (inferoanterior BA 40) | −63 | −27 | 18 | 5.160 | 1.095 | 0.027 |
| 7 | Left 43 | Area 43 | −57 | 3 | 3 | 4.927 | 1.114 | 0.029 |
| 8 | Right 6r | Rostral area 6 | 57 | 6 | 0 | 4.916 | 1.184 | 0.031 |
| 9 | Right PGi | PGi (inferior BA 39) | 54 | −60 | 30 | 4.739 | 1.007 | 0.027 |
| 10 | Right 8BL | Area 8B lateral | 21 | 36 | 51 | 4.655 | 0.713 | 0.020 |
| 11 | Right 7PC | Area 7PC | 36 | −45 | 54 | 4.414 | 0.712 | 0.021 |
| 12 | Left 9p | Area 9 posterior | −15 | 45 | 45 | 4.199 | 0.639 | 0.019 |
| 13 | Right 6v | Ventral area 6 | 54 | 9 | 33 | 4.037 | 0.766 | 0.024 |
| 14 | Left 8Av | Ventral area 8A | −39 | 18 | 48 | 3.990 | 0.561 | 0.018 |
| 15 | Left AIP | Anterior intra-parietal area | −33 | −45 | 45 | 3.474 | 0.567 | 0.021 |
| 16 | Right FST | Fundus of the superior temporal area | 54 | −60 | 0 | 3.156 | 0.729 | 0.030 |
| 17 | Right 9m | Area 9 middle | 3 | 54 | 24 | 3.128 | 0.609 | 0.025 |
| 18 | Left 31pv | Area 31p ventral | −3 | −51 | 33 | 3.049 | 0.784 | 0.033 |
| 19 | Right VIP | Ventral intra-parietal complex | 18 | -63 | 57 | 2.984 | 0.572 | 0.025 |
| 20 | Right 6a | Area 6 anterior | 33 | 3 | 63 | 2.975 | 0.454 | 0.020 |
| 21 | Left FOP4 | Frontal opercular area 4 | −33 | 21 | 6 | 2.858 | 0.828 | 0.037 |
| 22 | Right 5mv | Area 5m ventral | 12 | −30 | 45 | 2.822 | 0.701 | 0.032 |
| 23 | Right 46 | Area 46 | 36 | 42 | 30 | 2.779 | 0.656 | 0.030 |
| 24 | Left 10v | Area 10v | 0 | 57 | −9 | 2.769 | 0.731 | 0.034 |
| 25 | Left p9-46v | Area posterior 9-46v | −42 | 36 | 27 | 2.591 | 0.561 | 0.028 |
| 26 | Left V3A | Area V3A | −15 | −90 | 33 | 2.575 | 0.684 | 0.034 |
| 27 | Left TE1a | Area TE1 anterior | −63 | −15 | −15 | 2.527 | 0.595 | 0.030 |
| 28 | Right TE1a | Area TE1 anterior | 60 | −9 | −21 | 2.494 | 0.580 | 0.030 |
| 29 | Right IFSa | Anterior inferior frontal suleus | 48 | 39 | 12 | 2.468 | 0.480 | 0.025 |
| 30 | Left 7Am | Medial area 7A | −12 | −60 | 60 | 2.461 | 0.475 | 0.025 |
| 31 | Right V3A | Area V3A | 18 | −87 | 36 | 2.442 | 0.645 | 0.034 |
| 32 | Right V4 | Fourth visual area | 24 | −63 | −9 | 2.339 | 0.446 | 0.024 |
| 33 | Left 6a | Area 6 anterior | −24 | 3 | 63 | 2.317 | 0.331 | 0.018 |
| 34 | Left VMV1 | Ventromedial visual area 1 | −18 | −60 | −6 | 1.937 | 0.397 | 0.026 |
| 35 | Left FEF | Frontal eye fields | −45 | −9 | 57 | 1.412 | 0.640 | 0.058 |

*BA* Brodmann area.

cohort number (Fig. 1c). According to prior brain network parcellations[29,30], these identified hub voxels (15,461 voxels) were spatially distributed in multiple brain networks, including the DMN (27.5%), dorsal attention network (DAN) (16.5%), FPN (15.9%), ventral attention network (VAN) (15.6%), somatomotor network (SMN) (14.4%), and visual network (VIS) (9.9%) (Fig. 1d). Using a local maxima localization procedure, we identified 35 robust brain hubs across 61 cohorts (Fig. 1c and Table 1), involving various heteromodal and unimodal areas. Specifically, the most robust findings resided in several lateral parietal regions, including the bilateral ventral postcentral gyrus, supramarginal gyrus, and angular gyrus.

**The identified connectome hubs are reproducible across cohorts and individuals**. During identifying the above highly consistent connectome hubs, the random-effects meta-analysis revealed high heterogeneity of FCS across cohorts (Fig. 2a). The cumulative distribution function plot shows more than 95% voxels with $I^2$ (heterogeneity score) exceeding 50% (Fig. 2b), indicating high heterogeneity across cohorts in almost all brain areas (see also Supplementary Fig. 1). To determine whether the connectome hubs identified here are dominated by certain cohorts or are reproducible across cohorts and individuals, we performed a leave-one-cohort-out validation analysis and an across-subject/cohort conjunction analysis.

*Leave-one-cohort-out validation analysis*. We repeated the above harmonized meta-analysis hub identification procedure after leaving one cohort out at a time. Comparing the identified hubs using all cohorts (Fig. 1c) with those after leaving one cohort out obtained extremely high Dice's coefficients (*mean ± sd*: 0.990 ± 0.006; range: 0.966-0.997). For hub peaks, leaving one cohort out resulted in very few displacements (mostly fewer than 6 mm, Fig. 2c, d). Thus, connectome hubs identified using the 61 cohorts were not dominated by specific cohorts.

*Across-subject/cohort conjunction analysis*. We defined the top $N$ ($N = 15,461$, which is the voxel number of hubs in Fig. 1c) voxels with the highest FCS values of a subject or a cohort as connectome hubs for that subject or that cohort. Then, for each voxel, we assessed hub occurrence probability values across subjects and cohorts. The identified hubs using all cohorts were highly overlapped with the top $N$ voxels with the highest hub occurrence probability values both across all subjects and across all cohorts, indicated by a high Dice's coefficient (*Dice* = 0.867, Fig. 2e; *Dice* = 0.924, Fig. 2f). When the identified hubs using all cohorts were compared with the top $N$ voxels with the highest hub occurrence probability values across randomly selected subjects or across randomly selected cohorts, the Dice's coefficient approached 99% of its maximum value after exceeding 510 subjects (Fig. 2g) and 35 cohorts (Fig. 2h), respectively. This

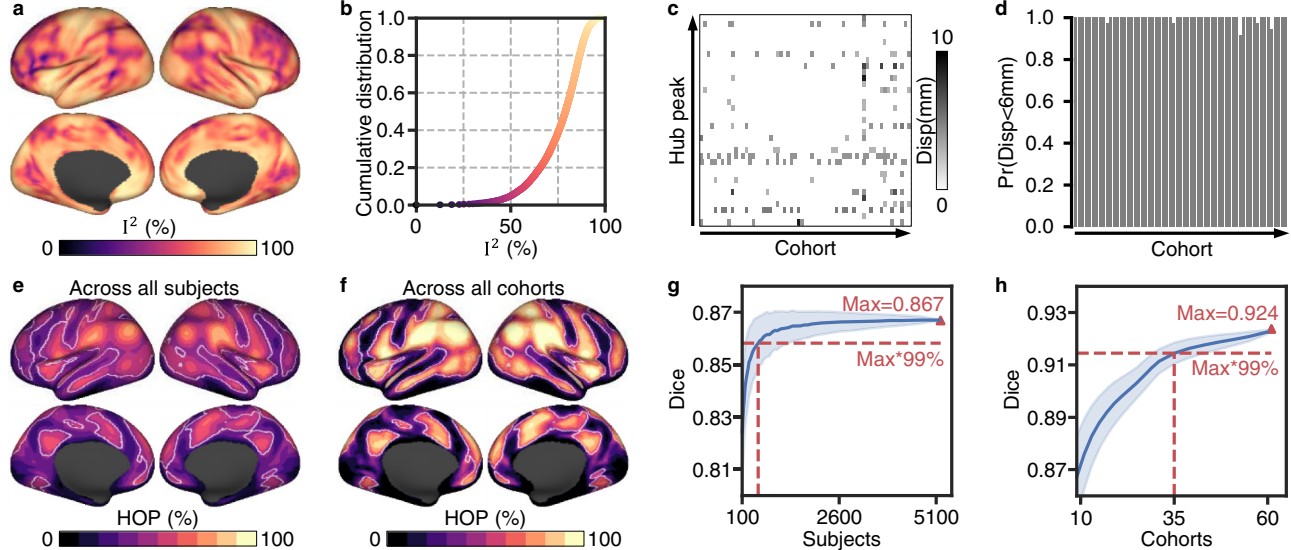

**Fig. 2 The identified connectome hubs are reproducible across cohorts and individuals. a** Heterogeneity measurement $I^2$ estimated through the random-effects meta-analysis. **b** Cumulative distribution function plot of $I^2$. **c** Heatmap of displacements of the 35 hub peaks after leaving one cohort out. **d** Bar plot of the probability across the 35 hub peaks whose displacement was less than 6 mm after leaving one cohort out. **e, f** Hub occurrence probability (HOP) map across all subjects (**e**) and all cohorts (**f**). White lines delineate boundaries of the identified hubs in Fig. 1c. **g, h** Dice's coefficient of the identified hubs in Fig. 1c compared with the top $N$ (voxel number of the identified hubs in Fig. 1c) voxels with the highest hub occurrence probability values across randomly selected subjects (**g**) and randomly selected cohorts (**h**). Blue shading represents the standard deviation across 2000 random selections.

indicated that the identified connectome hubs were highly reproducible both across cohorts and across individuals.

Validation analysis demonstrated that the above results did not depend on analysis parameters, such as the connection threshold (Supplementary Figs. 2 and 3), and were not driven by the size of the brain network to which they belong[31] (Supplementary Fig. 4), suggesting the robustness of our main findings.

**Connectome hubs have heterogeneous functional connectivity profiles.** Next, we further examined whether these robust brain hubs (Fig. 1c and Table 1) have distinctive functional connectivity profiles that represent their unique roles in network communication. To gain detailed and robust functional connectivity profiles of each hub region, we conducted a seed-to-whole-brain connectivity meta-analysis in a harmonized protocol again. For each of the 35 hub regions, we obtained an estimated Cohen's $d$ effect size map that characterizes the robust whole-brain connectivity pattern relevant to the seed region across the 61 cohorts (Fig. 3). We then divided the connectivity map of each hub into eight brain networks according to prior parcellations[29,30], resulting in an 8×35 connectivity matrix with each column representing the voxel percentage of each of the eight networks connected with a hub.

Hierarchical clustering analysis on the connectivity matrix clearly divided the 35 hubs into three clusters (Fig. 4a, b). Cluster I consists of 21 hubs that are primarily connected with extensive areas in the DAN, VAN, FPN, and SMN (orange, Fig. 4c). Cluster II consists of four hubs that are densely connected with VIS (green, Fig. 4c). Cluster III consists of 10 hubs that have robust connections with the DMN and LIMB (blue, Fig. 4c). Of particular interest is that within Cluster III, a left posterior middle frontal hub called ventral area 8A (8Av) shows a distinctive connectivity profile in contrast to the other nine hubs, manifested as having robust connections with bilateral frontal FPN regions (Fig. 3 and Supplementary Fig. 5). This implies that the left 8Av hub is a key connector between the DMN and FPN, which can be supported by the recent finding of a control-default connector located in the posterior middle frontal gyrus[32].

Although both Cluster I and III hubs are connected with subcortical structure (Fig. 4c), they are connected with different subcortical nuclei (Supplementary Fig. 6). Finally, whereas all hubs possess dense intranetwork connections, most also retain substantial internetwork connections (Supplementary Fig. 7), which preserves efficient communication across the whole brain network feasible.

**Transcriptomic data distinguishes connectome hubs from non-hubs.** A supervised machine learning classifier based on XGBoost[33] and 10,027 genes' transcriptomic data from the AHBA[34] was trained to distinguish connectome hubs from non-hubs (Fig. 5a). The sensitivity, specificity, and accuracy rate of the XGBoost classifier were stably estimated by repeating the training and testing procedure 1000 times. This classifier performed better than chance in all 1000 repetitions and achieved an overall accuracy rate of 65.3% (Fig. 5b). In cross-validation, connectome hubs and non-hubs were classified with a sensitivity of 71.1% and specificity of 63.4%, respectively. The testing procedure yielded a comparable sensitivity of 69.7% and specificity of 62.0%. After training the classifier, each gene's contribution to the optimal prediction model was determined. We noted that some key genes contributed two or three orders of magnitude more than other genes (Fig. 5c and Supplementary Data 1). The contributions of the top 300 genes with the greatest contributions to the XGBoost classifier were consistent between the first 500 repetitions and the second 500 repetitions (Pearson's $r = 0.958$, $p < 10^{-6}$, Fig. 5d), suggesting a high reproducibility.

To exclude the XGBoost model's potential bias relating to the mostly contributed key genes, we replicated the above classification results using another machine learning model based on the support vector machine (SVM) that was trained using only the top $N$ key genes with the greatest contributions to the XGBoost classifier (Fig. 5e). Because no data were available to determine how many key genes were sufficient to train an SVM classifier, we examined the count $N$ from 100 to 300. The SVM classifier achieved a very high peak accuracy rate of 91.8% with approximately the top 150 key genes in the easiest classification

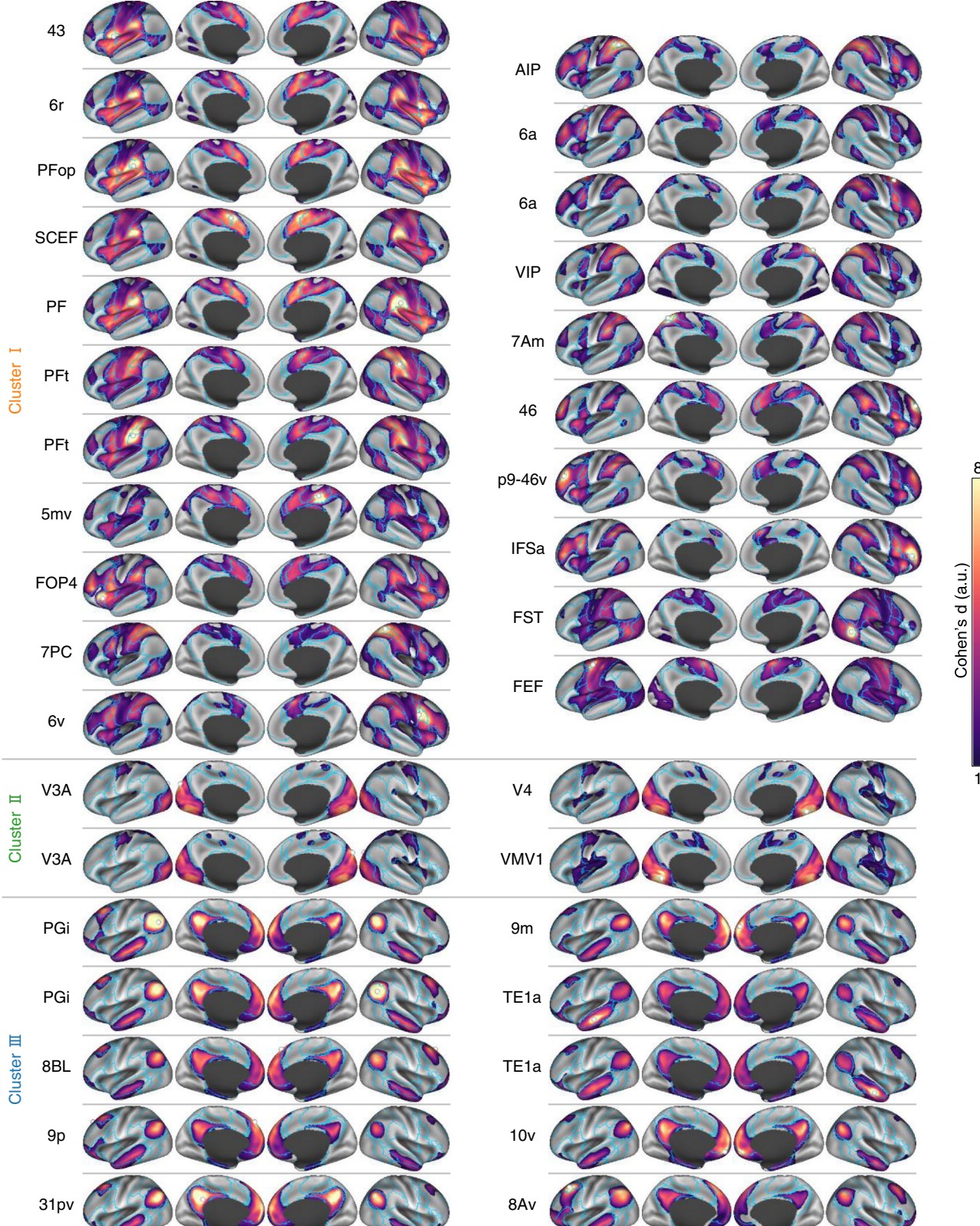

**Fig. 3 Functional connectivity maps of connectome hubs.** Cluster labels were derived by the hierarchical clustering solution in Fig. 4a. White spheres represent hub seeds. Blue lines delineate boundaries of the seven cortical networks shown in Fig. 1d. a.u. arbitrary unit.

task (Fig. 5f) and also achieved a reasonable peak accuracy rate of 67.8% with approximately the top 150 key genes even in the most difficult classification task (Fig. 5g). By contrast, SVM classifiers trained using 150 randomly selected genes performed worse than that using the top 150 key genes in all 1000 repetitions (Fig. 5h).

Validation analyses showed that the XGBoost and SVM classifiers trained using surrogate hub identification maps with

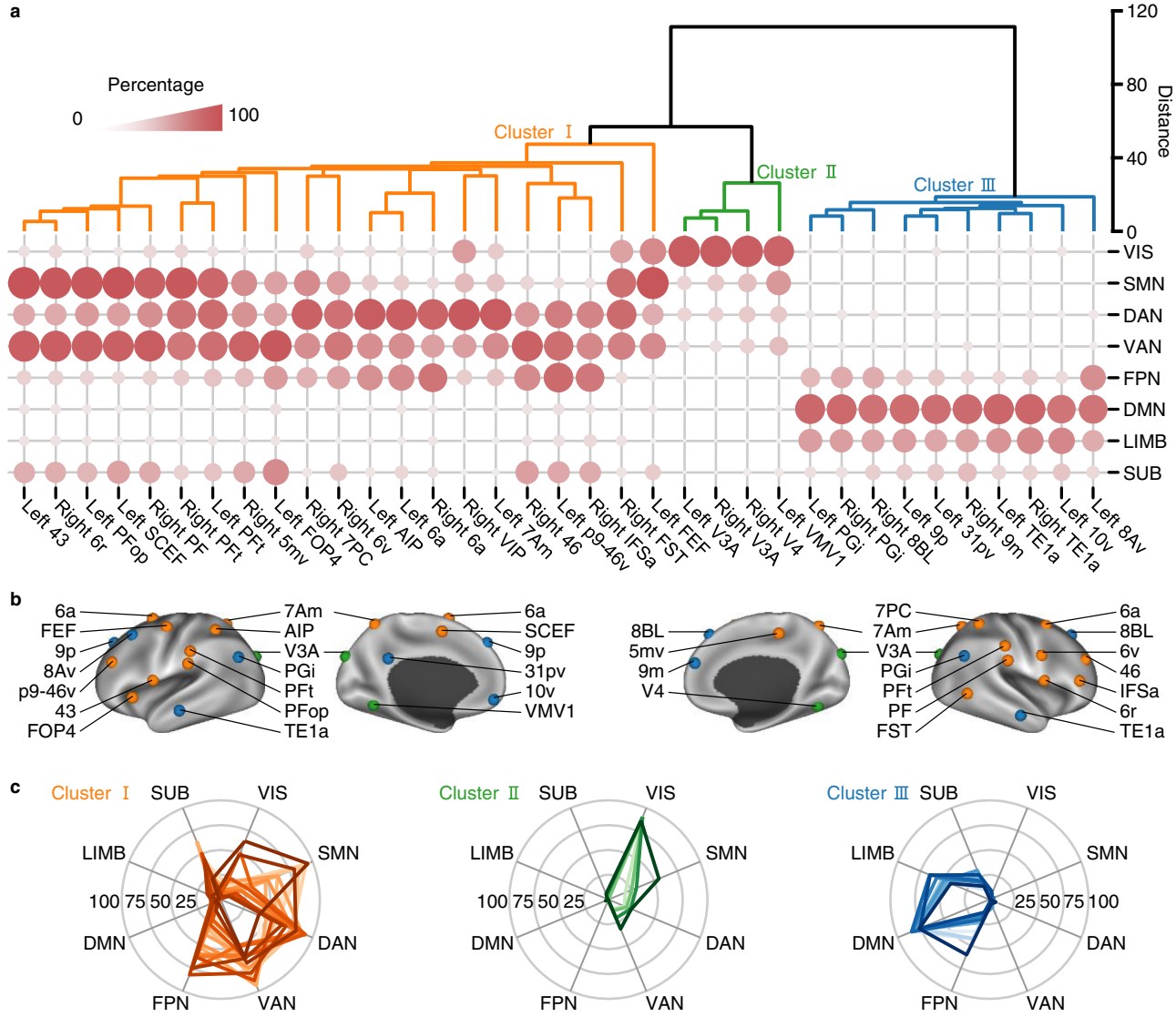

**Fig. 4 Hierarchical clustering analysis on connectome hubs' functional connectivity maps. a** Dendrogram derived by hierarchical clustering on the connectivity percentage matrix. **b** The 35 hubs were rendered using three different colors according to the hierarchical clustering solution. **c** Radar charts showing heterogeneous connectivity profiles of the three hub clusters.

the spatial autocorrelations being corrected performed no better than the chance level (Supplementary Fig. 8), confirming that the performance of the XGBoost and SVM classifiers was not driven by the effects of spatial autocorrelation inherent to the hub localization and the transcriptomic data. Thus, these robust connectome hubs were apparently associated with a transcriptomic pattern dominated by approximately 150 key genes.

**Connectome hubs have a spatiotemporally distinctive transcriptomic pattern.** Gene Ontology (GO) enrichment analysis using GOrilla[35] demonstrated that the above 150 key genes were mostly enriched in the neuropeptide signaling pathway (*fold enrichment* (FE) = 8.9, uncorrected $p = 1.2 \times 10^{-5}$, Supplementary Data 2). GO enrichment analysis using the ranked 10,027 genes according to their contributions to the XGBoost classifier also confirmed the most enriched GO term of the neuropeptide signaling pathway (FE = 5.7, uncorrected $p < 10^{-6}$, Supplementary Data 3). The ranked 10,027 genes were also associated with the developmental process (FE = 1.2), cellular developmental process (FE = 1.3), anatomical structure development (FE = 1.3), and neuron projection arborization (FE = 13.7) (uncorrected

$ps < 5.5 \times 10^{-4}$, Supplementary Data 3). We speculated that connectome hubs have a distinctive transcriptomic pattern of neurodevelopmental processes in contrast to non-hubs.

We repeated the GO enrichment analysis of the above 150 key genes using DAVID[36,37] and confirmed the mostly enriched GO term of the neuropeptide signaling pathway (FE = 8.7, uncorrected $p = 5.8 \times 10^{-4}$, Supplementary Data 4). In addition, there were 10 GO terms associated with metabolic process, such as the positive regulation of cellular metabolic process (FE = 1.4, uncorrected $p = 0.031$, Supplementary Data 4). Disease association analysis demonstrated metabolic disease associated with the greatest number of key genes (60 genes, FE = 1.2, uncorrected $p = 0.094$, Supplementary Data 5). Accordingly, it is rational to speculate that connectome hubs have a distinctive transcriptomic pattern of metabolic processes in contrast to non-hubs.

To confirm the above two speculations of GO enrichment analysis results, we examined transcription level differences between hub and non-hub regions for genes previously implicated in key neurodevelopmental processes[38] (Supplementary Data 6) and main neuronal metabolic pathways[39] (oxidative phosphorylation[40] and aerobic glycolysis[41], Supplementary Data 7). Permutation tests

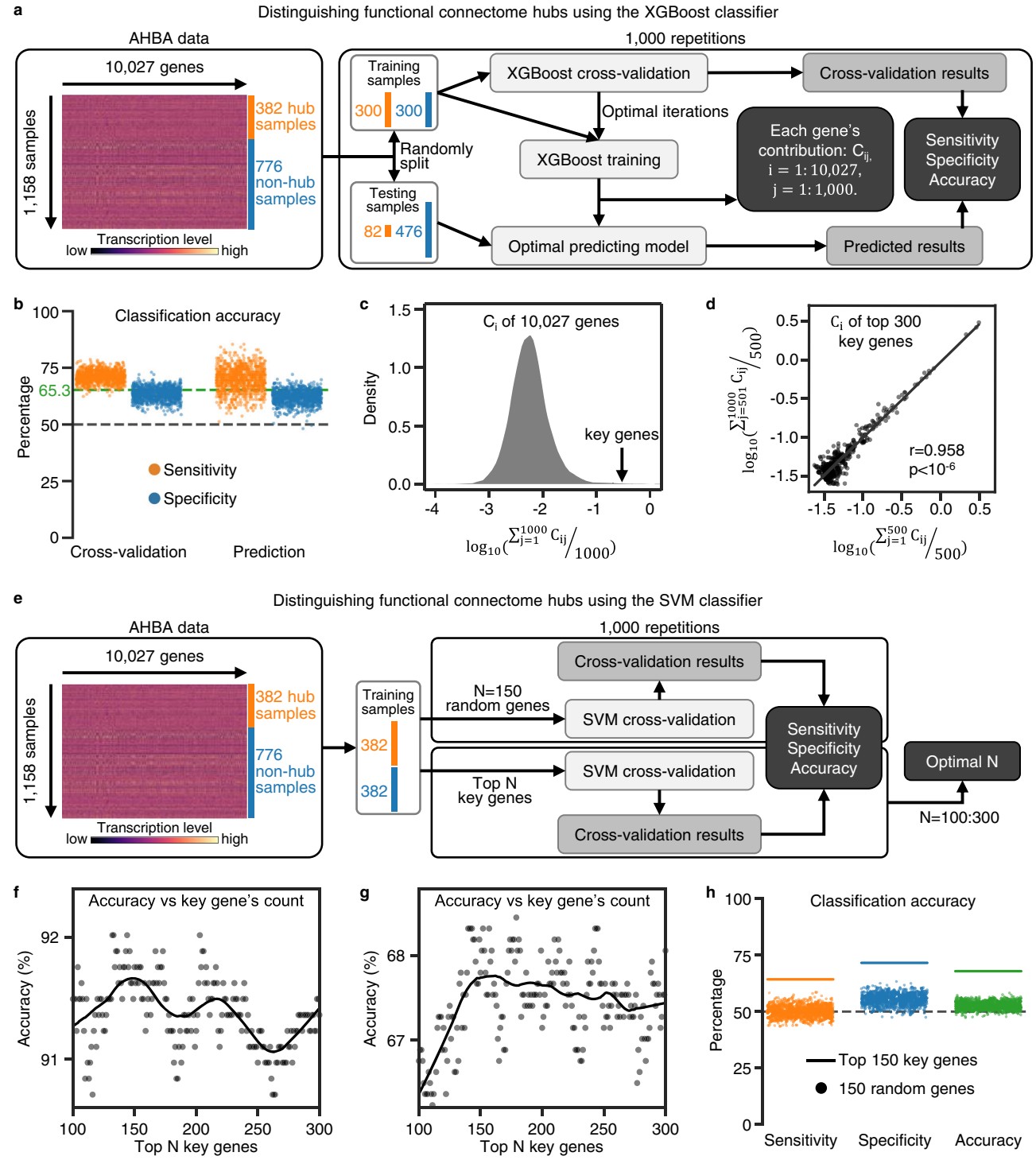

**Fig. 5 Transcriptomic data distinguishes connectome hubs from non-hubs. a** Schematic diagram of using the XGBoost model to classify brain samples as a hub or non-hub. **b** Performance of the XGBoost classifier. Each dot represents one repetition in **a**. The horizontal gray dashed line represents the chance level accuracy rate (50%). The horizontal green dashed line represents the average accuracy rate (65.3%) of the XGBoost classifier across 1000 repetitions. **c** Density plot of 10,027 genes' logarithmic average contributions across 1000 repetitions to the XGBoost classifier. Genes with the greatest contributions were regarded as key genes. **d** Regression plot of the logarithmic average contributions of the top 300 key genes across the first 500 repetitions versus those across the second 500 repetitions. Each dot represents one gene. **e** Schematic diagram of using the SVM model to classify brain samples as a hub or non-hub. **f**, **g** Accuracy rate of the SVM classifier versus the count of key genes used to distinguish 382 hub samples from 382 non-hub samples with the highest rate (**f**) or lowest rate (**g**) to be correctly classified by the XGBoost classifier. Each dot represents one SVM classifier. Black curves were estimated by locally weighted regression. **h** Performance of the SVM classifier. Horizontal lines correspond to the SVM classifier trained using top 150 key genes in **g**. Each dot represents one repetition using 150 randomly selected genes in **e**. The horizontal gray dashed line represents the chance level accuracy rate (50%).

revealed hub regions with significantly higher transcription levels for genes associated with dendrite development, synapse development, and aerobic glycolysis than non-hub regions (one-sided Wilcoxon rank-sum tests, Bonferroni-corrected $ps \leq 0.032$, Fig. 6a). In addition, hub regions had a weak trend of lower transcription levels for genes associated with axon development, myelination, and neuron migration, and a higher transcription level for genes associated with oxidative phosphorylation (Fig. 6a). These differences in transcription level were consistent with our speculations of GO enrichment analysis results.

These above transcriptomic results were derived from the AHBA, an adult transcriptomic dataset. To explore their developmental evolutions, we inspected the developmental trajectory of transcription level in hub and non-hub regions respectively using the BrainSpan Atlas[42]. We observed diverging developmental trajectories of transcription level between hub and non-hub regions in these key neurodevelopmental processes and main neuronal metabolic pathways (Fig. 6b and Supplementary Fig. 9a). The magnitude of differences in developmental trajectory between hub and non-hub regions continuously exceeds the median absolute deviation of transcription level across brain regions during some periods (Fig. 6c and Supplementary Fig. 9b), suggesting a trend of greater difference than expected. Specifically, hub regions have higher transcription levels for neuron migration during the late-fetal period, higher transcription levels for dendrite and synapse development from the late-childhood to mid-adolescence period, and lower transcription levels for axon development and myelination from the mid-childhood to late-adolescence period than non-hub regions. These results are in agreement with the observation of primary somatosensory, auditory, and visual (V1/V2) cortices with lower synapse density but higher myelination than the prefrontal area[43,44]. Moreover, hub regions have higher transcription levels than non-hub regions for aerobic glycolysis since the early childhood period. These transcriptome analyses achieved convergent results between the AHBA and BrainSpan Atlas.

Together, functional connectome hubs have a spatiotemporally distinctive transcriptomic pattern in contrast to non-hubs, which is dominated by genes involved in the neuropeptide signaling pathway, neurodevelopmental processes, and metabolic processes.

**Neural contextualization of connectome hubs' transcriptomic pattern.** To assess the neural relevance of the above identified transcriptomic pattern underlying functional connectome hubs, we contextualized it relative to prior established neuroimaging maps. The identified transcriptomic pattern is dominated by genes with the highest enrichment for the neuropeptide signaling pathway. Considering that neuropeptides are a main type of indirect neurotransmitter widely distributed in the human central nervous system and their vital role in modulating direct excitatory and inhibitory transmission[45], it is rational to speculate that there are apparent differences in neurotransmitter systems between hub and non-hub regions. Using neurotransmitter maps derived from positron emission tomography and single photon emission computed tomography[46], we found that hub regions have higher density of GABAa, glutamate, mu opiod, cannabinoid, dopamine D2, and serotonin receptor and norepinephrine transporter but lower density of dopamine transporter and fluorodopa than non-hub regions (one-sided Wilcoxon rank-sum tests, Bonferroni-corrected $ps \leq 0.015$, Fig. 7a).

Growing evidence has suggested a striking spatial correspondence between transcriptomic profile and structural connectivity in the human brain[27]. We speculated that the above differences in microscale transcriptome between hub and non-hub regions in

key neurodevelopmental processes may result in differences in macroscale structural connectivity profile. Using a fiber length profiling dataset[47], we observed that hub regions possess more fibers with a length exceeding 40 mm but less fibers with a length shoter than 40 mm (one-sided Wilcoxon rank-sum tests, Bonferroni-corrected $ps \leq 0.007$, Fig. 7b), suggesting a more intricate fiber configuration in hub regions.

The above transcriptome analyses have shown a higher transcription level of oxidative phosphorylation and aerobic glycolysis in hub regions than in non-hubs. We validated this observation using a metabolism dataset derived from positron emission tomography[48] and found that hub regions not only have a higher metabolic rate than non-hubs in oxidative phosphorylation (indicated by the cerebral metabolic rate for oxygen) and aerobic glycolysis (indicated by the glycolytic index), but also have more blood supply (indicated by the cerebral blood flow) (one-sided Wilcoxon rank-sum tests, Bonferroni-corrected $ps < 0.001$, Fig. 7c). This is in agreement with prior observations of a tight coupling between FCS and blood supply[1,49].

In addition, we also noted that the above 150 key genes are enriched for several psychiatric disorders ($FE = 3.5$, uncorrected $p = 5.5 \times 10^{-4}$, Supplementary Data 5). This finding is in accordance with prior observations of hub regions being preferentially targeted by neuropsychiatric disorders[5–8]. This implies that connectome hubs may have different susceptibility to neuropsychiatric disorders in contrast to non-hubs. We validated it by performing an association analysis between the effect size of connectome hub and the effect size of cortical thickness atrophy in neuropsychiatric disorders[50]. We observed that the Cohen's $d$ of connectome hub is negatively correlated with the Cohen's $d$ of cortical thickness atrophy in 22q deletion syndrome (Pearson's $r = -0.292$, uncorrected $p = 0.009$) and autism spectrum disorder (Pearson's $r = -0.333$, uncorrected $p = 0.019$) but positively correlated with the Cohen's $d$ of cortical thickness atrophy in bipolar disorder (Pearson's $r = 0.418$, uncorrected $p = 0.003$) and schizophrenia (Pearson's $r = 0.247$, uncorrected $p = 0.040$) (Fig. 7d). This suggests that connectome hubs have a trend of higher susceptibility to cortical thickness atrophy in bipolar disorder and schizophrenia but lower susceptibility to cortical thickness atrophy in 22q deletion syndrome and autism spectrum disorder than non-hubs.

## Discussion

Using a worldwide harmonized meta-connectomic analysis of 5212 healthy young adults across 61 cohorts, we provided, to the best of our knowledge, the first description of highly consistent and reproducible functional connectome hubs in the resting human brain. Using transcriptomic data from the AHBA and BrainSpan Atlas, we reported that these robust connectome hubs have a spatiotemporally distinctive transcriptomic pattern in contrast to non-hub regions. These results advanced our knowledge of the robustness of macroscopic functional connectome hubs and their potential cellular and molecular substrates.

Extant reports have shown largely inconsistent and less reproducible hub localizations[7,8,13–19], which may arise from high heterogeneity in the included subjects, data acquisition, and analysis strategies across studies. To diminish these potential confounding factors, we employed stringent participant inclusion criteria that included only healthy young adults aged 18–36 years and adopted harmonized data preprocessing and connectome analysis protocols across cohorts. Nevertheless, the random-effects meta-analysis revealed high heterogeneity among cohorts in almost all brain areas, which implied that heterogeneity of imaging scanners and/or imaging protocols could be an important cause for inconsistent and less reproducible results across

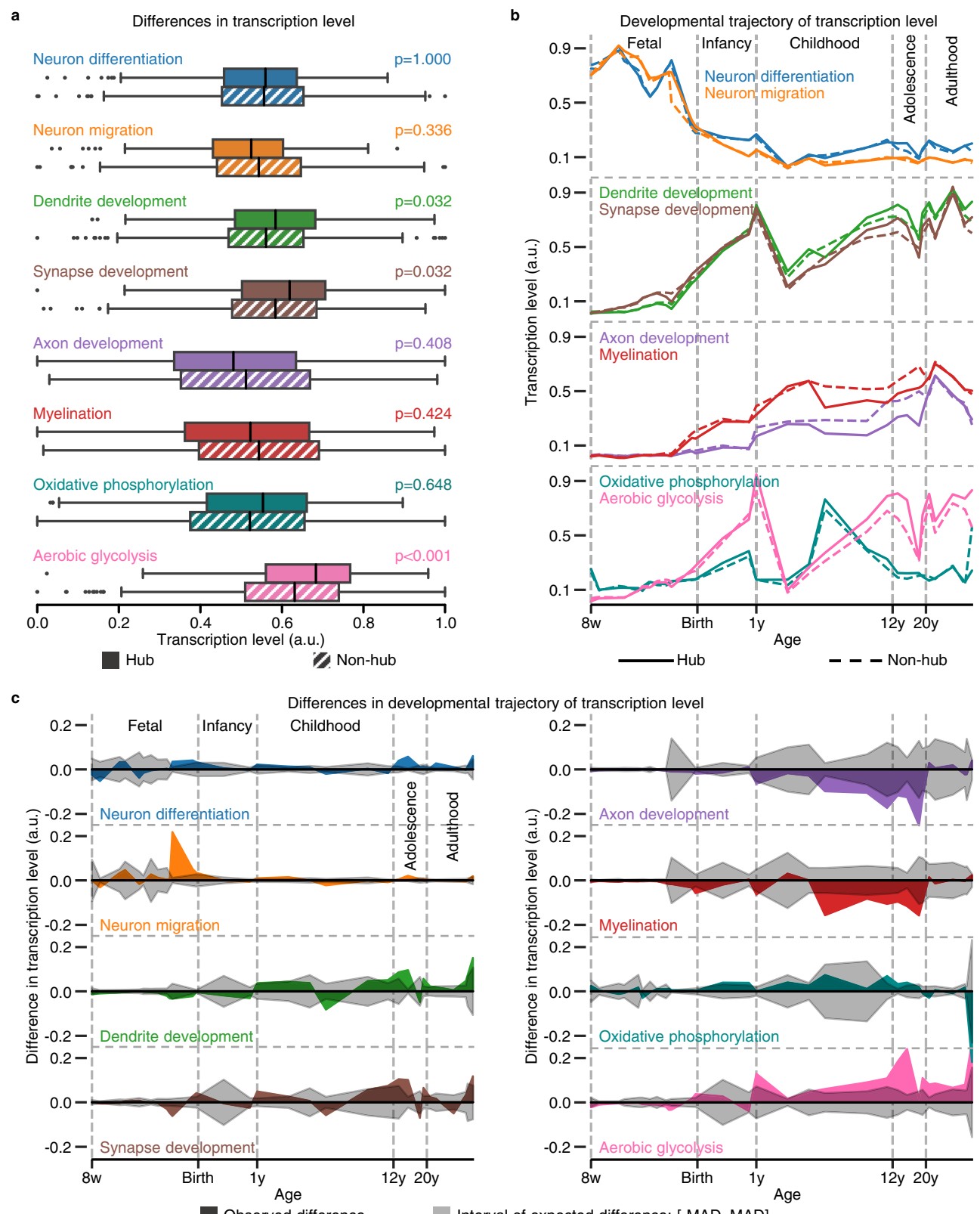

**Fig. 6 Connectome hubs have a spatiotemporally distinctive transcriptomic pattern. a** Differences in transcription level between hub samples ($n = 382$) and non-hub samples ($n = 776$) for genes associated with key neurodevelopmental processes[38] and main neuronal metabolic pathways[39]. Boxplot left and right edges, vertical black lines, and whiskers and dots depict the 25th and 75th percentiles, median, and extreme nonoutlier and outlier values, respectively. The statistical significances of one-sided Wilcoxon rank-sum tests were determined by 1000 permutation tests and were labeled with Bonferroni-corrected *p* values. **b** Developmental trajectory of transcription level in hub and non-hub regions for genes involved in key neurodevelopmental processe[38] and main neuronal metabolic pathways[39]. **c** Differences in the developmental trajectory of transcription level between hub and non-hub regions shown in **b**. MAD, the median absolute deviation of transcription level across brain regions. w post-conceptional week, y postnatal year, a.u. arbitrary unit.

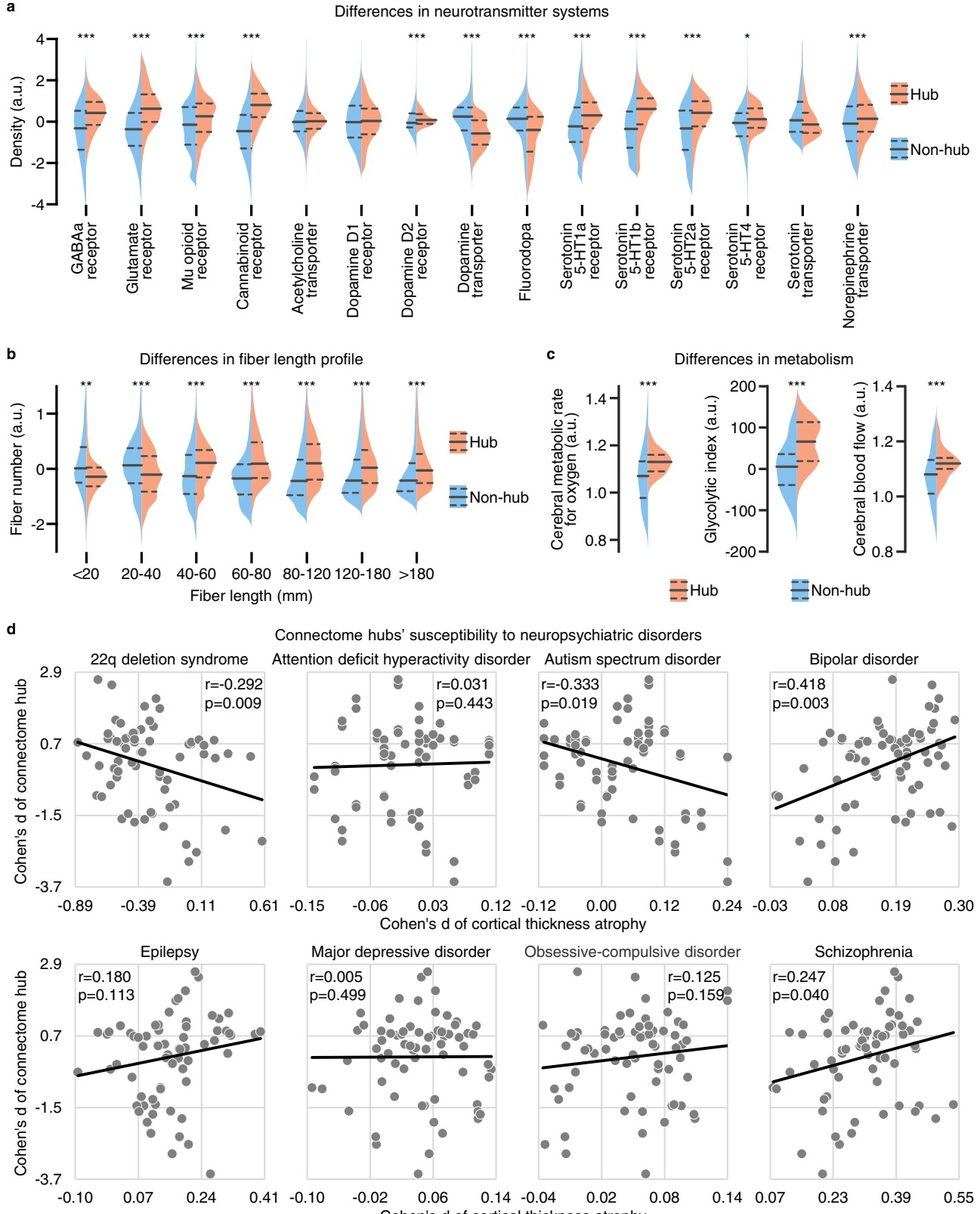

prior studies. Thus, our study was indispensable by conducting a harmonized random-effects meta-analysis model in which both intracohort variation (i.e., sampling errors) and intercohort heterogeneity were considered[51]. In addition, our validation results showed that the spatial distribution of functional connectome hubs was relatively stable when using more than 510 subjects and

35 cohorts, demonstrating that 5212 subjects from 61 cohorts were adequate to minimize the effects of both sampling errors and heterogeneity among cohorts. Considering only dozens of subjects in most prior studies[7,8,13–15,17,19], the low statistical power attributed to inadequate subjects could be another cause for prior inconsistent and less reproducible hub localizations.

**Fig. 7 Neural contextualization of connectome hubs' transcriptomic pattern. a–c** Differences between hub (red) and non-hub (blue) regions in density of neurotransmitter receptor and transporter (**a**, hub voxels $n = 15,461$, non-hub voxels $n = 32,158$), fiber number for different fiber length bins (**b**, hub vertices $n = 25,944$, non-hub vertices $n = 33,195$), and metabolic rate for oxygen, aerobic glycolysis, and blood supply (**c**, hub regions $n = 29$, non-hub regions $n = 60$). For each violin plot, dashed gray lines depict the 25th and 75th percentiles, solid gray line depicts median value. The statistical significances of one-sided Wilcoxon rank-sum tests were determined by 1000 permutation tests and were labeled with Bonferroni-corrected $p$ values. *$p < 0.05$, **$p < 0.01$, ***$p < 0.001$. a.u. arbitrary unit. **d** Regression plot of the Cohen's $d$ value of connectome hub versus the Cohen's $d$ value of cortical thickness atrophy across 68 cortical areas for eight disorders. Positive Cohen's $d$ value indicates thinning of cortical thickness in patients. Each dot represents one cortical area. The statistical significances of Pearson's correlation coefficients were determined by 1000 permutation tests and were labeled with uncorrected $p$ values.

Finally, we used harmonized image processing and connectome analysis protocols across cohorts, which avoided methodological variation and reduced potential methodological defects that have not been resolved in prior studies. See an extension discussion in Supplementary Note 1.

The present results demonstrated that the 35 highly consistent and reproducible connectome hubs show heterogeneous functional connectivity profiles, forming three clusters. Twenty-one hubs (Cluster I) are connected with extensive areas in the DAN, VAN, FPN, and SMN. Previous investigations indicated that they are core regions of the DAN (left AIP, right 7PC, left 7Am, bilateral PFt, left FEF, bilateral 6a, right 6v, and right FST)[29,52], VAN (left 43, left FOP4, right 46, right 6r, right PF, left PFop, left SCEF, right 5mv)[29,52], and FPN (left p9-46v and right IFSa)[29,53]. In addition, hub regions involved in the sensorimotor pathway (right VIP, right FST, left 7Am, and left FEF)[54] are also connected with the visual association cortex, acting as connectors between the VIS and the SMN, DAN, and VAN. Information flow along the primary visual, visual association, and higher-level sensorimotor cortices is undertaken by the four occipital hubs (Cluster II) left VMV1, right V4, and bilateral V3A that are all densely connected with the VIS and portions of the SMN, DAN, and VAN. This is supported by the report of their dense connections with both the visual system and SMN region the frontal eye field, DAN region the superior parietal cortex, and VAN region the parietal operculum and anterior insula[55] and also aligns with the role of their homologous regions in the non-human primate cerebral cortex[54]. The remaining 10 hubs (Cluster III) are all located in canonical DMN regions[56]. One of them, the left 8Av hub, is robustly connected with both DMN and lateral prefrontal FPN regions, acting as a connector between the DMN and FPN. This can be supported by the recent finding of a control-default connector located in the posterior middle frontal gyrus[32] and may also be a case of the hypothesis of parallel interdigitated subnetworks[57] where the posterior middle frontal gyrus is connected with a subnetwork of the DMN and some regions of the FPN. This observation offers a crucial complementary interpretation to the conventional assumption that the DMN is anticorrelated with other networks[56]. Considering that communication between the DMN and other networks is of particular relevance to neuropsychiatric disorders[58], such as autism spectrum disorders[59], we speculated that the left 8Av hub may be a promising target region for therapeutic interventions.

We demonstrated that these robust brain hubs have a spatiotemporally distinctive transcriptomic pattern dominated by genes with the highest enrichment for the neuropeptide signaling pathway. Because neuropeptides are a main type of indirect neurotransmitter that is widely distributed in the human central nervous system[45], robust neuropeptide signaling pathways are indispensable for efficient synaptic signal transduction that sustains dense and flexible functional connections of hub regions. This is also supported by our observation of differences in neurotransmitter receptor and transporter density between hub and non-hub regions. In addition, hub regions have higher transcription levels for main neuronal metabolic pathways in contrast

to non-hubs. This is reasonable because massive synaptic activities in hub regions demand high material and metabolic costs, which is in accordance with our observation of more blood supply and higher oxidative phosphorylation and aerobic glycolysis levels in hub regions. This is also consistent with prior observations of a tight coupling between FCS and blood supply[1,49].

We found that connectome hubs possess a spatiotemporally distinctive transcriptomic pattern of key neurodevelopmental processes in contrast to non-hubs. Specifically, connectome hubs have higher transcription levels for dendrite and synapse development and lower transcription levels for axon development and myelination during childhood, adolescence, and adulthood. These findings are compatible with previous observations of the prefrontal area having higher synapse density but lower myelination than primary somatosensory, auditory, and visual (V1/V2) cortices[43,44]. Higher transcription levels for dendrite and synapse development in hub regions are necessary for the overproduction of synapses that will be selectively eliminated based on the demand of the environment and gradually stabilized before full maturation[60], which has been proposed as the major mechanism of creating diverse neuronal connections beyond their genetic determination[60]. Lower transcription levels for axon development and myelination will prolong the myelination period in hub regions, which characterizes a delayed maturation phase[61]. Marked delay of anatomical maturation in human prefrontal and lateral parietal cortices has been frequently observed both in human development[62,63] and in primate evolution[61], which provides more opportunities for social learning to establish diverse neuronal circuits that contribute to our complex[63] and species-specific[61] cognitive capabilities. We also observed higher transcription levels for neuron migration in hub regions from mid-fetal period to early infancy. This is in agreement with the report of extensive migration of young neurons persisting for several months after birth in the human frontal cortex[64]. Meanwhile, the migration and final laminar positioning of postmitotic neurons are regulated by common transcription factors[65], which suggests that a higher transcription level for neuron migration in hub regions facilitates the construction of more intricate interlaminar connectivity. These microscale divergences of key neurodevelopmental processes may result in a more intricate macroscale structural connectivity profile in hub regions.

Human neurodevelopment is an intricate and protracted process, during which the transcriptome of the human brain requires precise spatiotemporal regulation[38]. Thus, in addition to contributing to our complex cognitive capabilities, the spatiotemporal differences in transcriptomic pattern of neurodevelopment between hub and non-hub regions may also increase brain connectome's susceptibility to neuropsychiatric disorders[61,63], which means small disturbance in the magnitude or the timing of this transcriptomic pattern may have long-term consequences on brain anatomical topography or functional activation. This is in line with the result of several psychiatric disorders being the most significant disease associated with the top 150 key genes and is also supported by our observation of differences in susceptibility

to cortical thickness atrophy in neuropsychiatric disorders between hub and non-hub regions. This implies that uncovering the intricate transcriptomic pattern, diverse neuronal circuits, anatomical topography, and functional activation of connectome hubs provide crucial and promising routes for understanding the pathophysiological mechanisms underlying neurodevelopmental disorders, such as autism spectrum disorders[38,59] and schizophrenia[5,38,61,63].

Of note, we conducted transcriptome–connectome association analysis using machine learning approaches in which non-linear mathematical operations were implemented rather than linear operations, such as linear correlation[24], linear regression[25], or partial least squares[26]. It has been argued that observations of transcriptome–connectome spatial association have a high false-positive rate through linear regression[66] and linear correlation[67] and may be largely shifted toward the first principal component axis of the dataset through partial least squares[68]. These investigations imply that prior transcriptome–connectome association results by linear mathematical operations may include high false-positive observations that are independent of connectome measurements, such as genes enriched for ion channels[24–26]. By contrast, high reproducibility across different machine learning models and across different GO enrichment analysis tools and convergent results from the AHBA and BrainSpan Atlas made it very unlikely that our findings were false-positive observations.

Some results of the present study should be interpreted cautiously because of methodological issues. First, we identified the robust connectome hubs using preprocessed rsfMRI data with global signal regression because of its great promise in minimizing physiological artifacts on functional connectomes[69]. Validation analysis demonstrated that hub distribution identified without global signal regression was more likely derived from physiological artifacts rather than by ongoing neuronal activity (Supplementary Note 2 and Supplementary Fig. 10). Second, we conducted a voxel-based connectome analysis in order to directly compare our results with the extant voxel-based reports[7,8,14–19] and increase the sensitivity of identifying spatially focal (e.g., voxel-sized) hubs[70]. The effects of parcellation-based[70] and surface-based[71] analysis on hub localizations should be resolved in future studies. Third, the AHBA dataset only includes partial human genes, of which approximately half were excluded in data preprocessing[34], which may have induced incomplete observations in our data-driven analysis. Finally, our transcriptomic signature results addressed only the association between connectome hubs and transcriptomic patterns and did not explore causation between them. Exploring more detailed mechanisms underlying this association is attractive and may be practicable for non-human primate brains in future studies.

## Methods

**Dataset**. We collected a large-sample rsfMRI dataset ($N = 7202$) from public data-sharing platforms and in-house cohorts, which consists of 73 cohorts from Asia, Europe, North America, and Australia. Data of each cohort were collected with participants' written informed consent and with approval by the respective local institutional review boards.

**Image preprocessing and quality control**. We first reviewed T1-weighted structural MRI data for all participants with the assistance of a neuroradiologist and a clinical neurologist to confirm no identifiable lesion or structural abnormality (e.g., regional atrophy and posterior cranial fossa arachnoid cyst). All rsfMRI data for the remaining participants were preprocessed routinely using SPM12 v6470 and GRETNA[72] v2.0.0 with a uniform pipeline. For each individual, we discarded the first 10 s' volumes for magnetic field stabilization and the participant's adaptation to the scanner. Next, slice-timing was corrected within each volume. To correct for head motion, all volumes were realigned to the mean image. Participants with significant head motion (translation above 3 mm or rotation above 3° in any direction) were excluded from the subsequent analyses. Then, all volumes were normalized to the 3-mm isotropic space of the Montreal Neurological Institute (MNI) using the EPI template provided by SPM12. The normalized

volumes were spatially smoothed using a 6-mm full-width at half-maximum Gaussian kernel. After that, the time series of each voxel underwent the procedure of linear trend removal, nuisance signal regression (24 head motion parameters, white matter, cerebrospinal fluid, and global brain signals), and temporal band-pass filtering (0.01–0.1 Hz). Finally, scrubbing was performed to minimize head motion effects[73]. Specifically, volumes with framewise displacement exceeding 0.5 mm and their adjacent volumes (1 back and 2 forward) were replaced with linearly inter-polated data. We excluded participants with more than 25% interpolated volumes. Notably, slice-timing was not corrected in the Human Connectome Project cohort due to multiband acquisition[74]. For participants with more than one rsfMRI scan, we used only one of them. To reduce the potential effects of development and aging on our results, we restricted our analysis to healthy young adults aged 18 to 36 years. To ensure sufficient statistical power, twelve cohorts were discarded due to having fewer than 10 participants that passed quality controls. After these stringent quality controls, we included preprocessed rsfMRI data of 5212 healthy young adults (2377 males) from 61 independent cohorts in the final analysis. The sample size and age ranges of each cohort were summarized in Fig. 8. Supplementary Data 8 provides detailed information on the individual cohorts.

**Identifying robust functional connectome hubs using a harmonized meta-analysis**. For each individual, we constructed a voxelwise functional connectome matrix by computing the Pearson's correlation coefficient between preprocessed rsfMRI time series of all pairs of voxels within a predefined gray matter mask (47,619 voxels). The gray matter mask was divided into seven large-scale cortical networks[29] and a subcortical network[30]. The cerebellum was not included due to largely incomplete coverage during rsfMRI scanning in most cohorts. Negative functional connections were excluded from our analysis due to neurobiologically ambiguous interpretations[75]. To further reduce the bias of signal noise and simultaneously avoid the effect of potential sharing signals between nearby voxels, both weak connections (Pearson's $r < 0.1$) and connections terminating within 20 mm were set to zero[76]. We validated the threshold of weak connections using 0.05 and 0.2 (Supplementary Figs. 2 and 3). For each voxel, we computed the FCS as the sum of connection weights between the given voxel and all the other voxels. We further normalized this resultant FCS map with respect to its mean and standard deviation across voxels[7].

For each cohort, we performed a general linear model on these normalized FCS maps to reduce age and sex effects. For each voxel, we constructed the general linear model as:

$$FCS_i = \beta_0 + \beta_{Age} * (Age_i - MeanAge) + \beta_{Sex} * Sex_i + \varepsilon_i \quad (1)$$

$FCS_i$, $Age_i$, $Sex_i$, and $\varepsilon_i$ indicate the FCS, age, sex, and residual of the $i$th individual, respectively. *MeanAge* indicates the mean age of that cohort. $\beta_0$ indicates the mean FCS of that cohort. The general linear model exported a mean FCS map and its corresponding variance map for each cohort.

The mean and variance FCS maps of the 61 cohorts were submitted to a random-effects meta-analysis model[51] to address across-cohort heterogeneity of functional connectomes. A short summary of the random-effects meta-analysis was provided in the following section. The detailed computational procedures are described in the book[51]. This resulted in a consistent FCS pattern (Fig. 1a) and its corresponding SE map (Fig. 1b). We compared the FCS of each voxel with the average of the whole brain (i.e., zero) using a $Z$ value and estimated effect size using Cohen's $d$ metric[51]:

$$Z = \frac{FCS - 0}{SE} \quad (2)$$

$$d = \frac{Z}{\sqrt{k}} \quad (3)$$

$k$ is the number of cohorts in the meta-analysis.

In line with a previous neuroimaging meta-analysis study[77], we performed 10,000 one-sided nonparametric permutation tests[28] to assign a $p$ value to the observed $Z$ value. For each iteration, after randomizing the spatial correspondence among cohorts' mean FCS maps (the spatial correspondence between a cohort's mean FCS map and its variance map was not changed), we repeated the computation procedure of the random-effects meta-analysis for each voxel and extracted the maximum $Z$ value of all voxels to construct a null distribution. A $p$ value was assigned to each voxel by comparing the observed $Z$ value to the null distribution. For a statistical significance level below 0.05, this $p$ value closely tracks the Bonferroni threshold[28].

Finally, we defined functional connectome hubs as brain regions with a $p$ value less than 0.001 and cluster size greater than 200 mm³ (Fig. 1c). The thresholds of $p$ value and cluster size were similar with the activation likelihood estimation algorithm[77]. We extracted MNI coordinates for each local peak $Z$ value terminating beyond 15 mm within each brain cluster using the *wb_command -volume-extrema* command in Connectome Workbench v1.4.2.

**Random-effects meta-analysis**. For each voxel, $M_i$, $SD_i$, and $N_i$ indicate the mean and standard division value and the participant number of the $i$th cohort, respectively. The original weight assigned to the $i$th cohort is the inverse of its

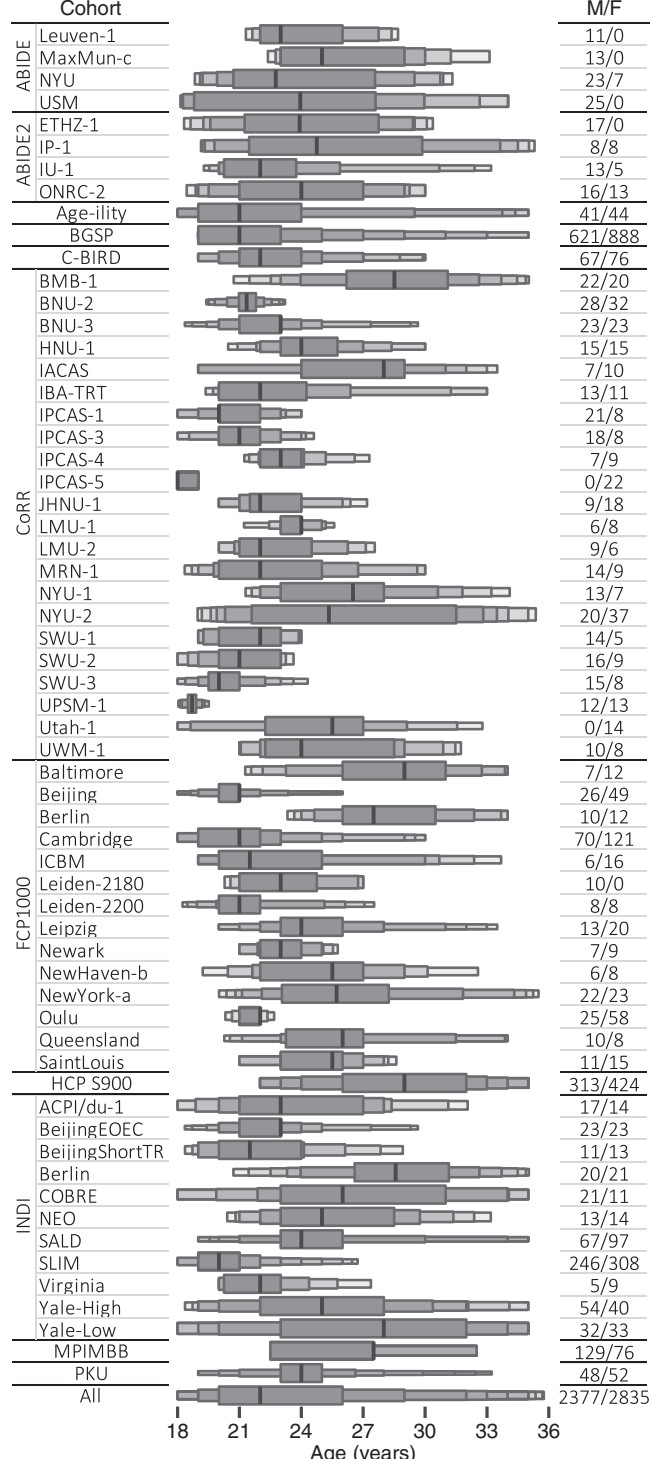

**Fig. 8 Enhanced box plot of the age ranges of each cohort.** Vertical black lines depict the median value. Left and right edges of the incrementally narrower boxes depict the lower and upper fourths, eighths, sixteenths, etc. M/F males/femals.

variance:

$$W_i = \frac{N_i}{SD_i^2} \tag{4}$$

The heterogeneity between cohort means was calculated as:

$$Q = \sum W_i M_i^2 - \frac{(\sum W_i M_i)^2}{\sum W_i} \tag{5}$$

The expected value of $Q$ is the degrees of freedom:

$$df = k - 1 \tag{6}$$

where $k$ is the number of cohorts in the meta-analysis. Therefore, the estimated variance of the cohort mean distribution was calculated as:

$$T^2 = \frac{Q - df}{\sum W_i - \frac{\sum W_i^2}{\sum W_i}} \tag{7}$$

The percentage of total variability that reflects heterogeneity among cohorts was calculated as:

$$I^2 = \frac{Q - df}{Q} \times 100\% \tag{8}$$

The weight assigned to the $i$th cohort was updated as:

$$W_i^* = \frac{1}{\frac{SD_i^2}{N_i} + T^2} \tag{9}$$

The result of the random-effects meta-analysis was calculated as:

$$M^* = \frac{\sum W_i^* M_i}{\sum W_i^*} \tag{10}$$

The variance of $M^*$ was estimated as:

$$V_{M^*} = \frac{1}{\sum W_i^*} \tag{11}$$

The standard error of $M^*$ was calculated as:

$$SE_{M^*} = \sqrt{V_{M^*}} \tag{12}$$

This random-effects meta-analysis model exported a mean value $M^*$, its corresponding standard error value $SE_{M^*}$, and the heterogeneity score $I^2$.

**Mapping seed-to-whole-brain connectivity maps of functional connectome hubs.** We modeled each hub seed region as a sphere with a 6-mm radius centered on the hub peak and computed Pearson's correlation coefficients between the seed region's preprocessed rsfMRI time series and the time series of all gray matter voxels. The time series of the seed region was computed by averaging the time series of all gray matter voxels in the seed sphere. These correlation coefficients were further transformed to Fisher's $z$ for normality.

For each cohort, we performed a general linear model on these Fisher's $z$ value maps to reduce age and sex effects. For each voxel, we constructed the general linear model as:

$$\text{Fisher's } z_i = \beta_0 + \beta_{Age} * (Age_i - MeanAge) + \beta_{Sex} * Sex_i + \varepsilon_i \tag{13}$$

Fisher's $z_i$, $Age_i$, $Sex_i$, and $\varepsilon_i$ indicate the Fisher's $z$, age, sex, and residual of the $i$th individual, respectively. $MeanAge$ indicates the mean age of that cohort. $\beta_0$ indicates the mean Fisher's $z$ value of that cohort. The general linear model exported a mean Fisher's $z$ value map and its corresponding variance map for each cohort.

The mean and variance Fisher's $z$ value maps of the 61 cohorts were submitted to the random-effects meta-analysis model[51] to address across-cohort heterogeneity of functional connections, resulting in a robust Fisher's $z$ pattern and its corresponding SE map. We compared the Fisher's $z$ value of each voxel with zero using a $Z$ value and estimated effect size using Cohen's $d$ metric[51]:

$$Z = \frac{\text{Fisher's } z - 0}{SE} \tag{14}$$

$$d = \frac{Z}{\sqrt{k}} \tag{15}$$

$k$ is the number of cohorts in the meta-analysis.

We performed 10,000 one-sided nonparametric permutation tests[28] to assign a $p$ value to the observed $Z$ value. For each iteration, after randomizing the spatial correspondence among cohorts' mean Fisher's $z$ value maps (the spatial correspondence between a cohort's mean Fisher's $z$ value map and its variance map was not changed), we repeated the computation procedure of the random-effects meta-analysis for each voxel and extracted the maximum $Z$ value of all voxels to construct a null distribution. Then, we assigned a $p$ value to each voxel by comparing the observed $Z$ value to the null distribution.

Finally, we defined the most consistent functional connectivity map as brain regions with a $p$ value less than 0.001 and cluster size greater than 200 mm³ (Fig. 3). To illustrate the left 8Av hub's connectivity map, we also mapped its contralateral region the right 8Av region's connectivity map (Supplementary Fig. 5).

**Hierarchical clustering analysis on connectivity maps of functional connectome hubs.** To address the effect of network size, we first divided the most consistent functional connectivity map of each hub into eight brain networks

mentioned above and represented the functional connectivity profile of a hub as the voxel percentage of each of the eight networks connected with it. Thus, we obtained an 8×35 connectivity matrix with each column representing the voxel percentage of each of the eight networks connected with a hub (Fig. 4a). Then, the 8×35 connectivity matrix was submitted to a hierarchical clustering model to obtain an agglomerative hierarchical cluster tree (Fig. 4a) that indicates the similarity of these functional connectivity profiles. We implemented hierarchical clustering model using the *linkage* function from MATLAB R2013a with default parameters.

**Identifying transcriptomic pattern underlying functional connectome hubs**. We trained machine learning classifiers based on XGBoost[33] and SVM to distinguish connectome hubs from non-hubs using transcriptomic features from the preprocessed AHBA dataset[34] (Fig. 5). The original AHBA dataset consists of microarray expression data of more than 20,000 genes in 3702 spatially distinct brain samples taken from six neurotypical adult donors[78]. Only two out of six donors were sampled from both hemispheres and the other four were sampled from only the left hemisphere. Because no statistically significant hemispheric difference was identified in the original AHBA dataset[78], a prior study[34] provided a publicly available preprocessed AHBA dataset that includes 10,027 genes' transcriptomic data from 1285 left cortical samples. Preprocessing steps taken by the study[34] mainly include probe-to-gene re-annotation, intensity based data filtering, probe selection, accounting for individual variability, and gene filtering. Of the 1,285 samples, 382 were identified as hub samples and 776 as non-hub samples according to their MNI coordinates and the hub identification map in Fig. 1c. The remaining 127 samples were not included in our analysis because they were out of our gray matter mask. The brain samples used in our analysis are listed in Supplementary Data 9.

We built a supervised machine learning classifier based on XGBoost, a scalable tree boosting system with state-of-the-art resource efficiency and superior performance in many machine learning challenges[33], to distinguish hub samples from non-hub samples using 10,027 genes' transcriptomic data from the preprocessed AHBA dataset. We used equal amounts of positive samples (hub samples) and negative samples (non-hub samples) in the classifier training procedure to ensure that the optimal classifier was unbiased toward any type of sample. To balance the time complexity and the prediction accuracy, we trained the classifier with 300 randomly selected hub samples and 300 randomly selected non-hub samples and tested it with the remaining 82 hub samples and 476 non-hub samples (Fig. 5a). Each gene's contribution to the optimal prediction model was determined after training the classifier. Based on previous experience[79], we performed a 30-fold cross-validation procedure to identify the optimal number of model training iterations. The sensitivity, specificity, and accuracy rate of the XGBoost classifier and each gene's contribution to the classification results were stably estimated by repeating the randomly selecting training samples, cross-validation, and classifier training and testing procedures 1000 times. We implemented XGBoost using the *XGBoost* package[33] v1.2.0.1 in R 4.0.2 with following parameters: $nrounds = 1500$, $early\_stopping\_rounds = 50$, $eta = 0.05$, $objective = $"binary:logistic".

To exclude the XGBoost model's potential bias relating to mostly contributed key genes, we reproduced classification results using another machine learning model based on SVM (Fig. 5e). Instead of using all 10,027 genes' transcriptomic features, we only used genes with the greatest contributions to the XGBoost classifier to train the SVM classifier. If the key genes with the greatest contributions to the classification results are independent of the XGBoost model, the SVM classifier will achieve a comparable or higher accuracy rate than the XGBoost classifier because of the exclusion of redundant genes whose contribution to the classification results is negligible. In line with the XGBoost model, we used equal amounts of hub samples and non-hub samples in the classifier training procedure. For the easiest classification task, the SVM classifier was trained to distinguish all 382 hub samples from 382 non-hub samples with the highest rate to be correctly classified by the XGBoost classifier. For the most difficult classification task, the SVM classifier was trained to distinguish all 382 hub samples from 382 non-hub samples with the lowest rate to be correctly classified by the XGBoost classifier. To balance the time complexity and the prediction accuracy, we performed a 382-fold cross-validation procedure to obtain the optimal SVM classifier. We implemented SVM using the *svm* function from the *scikit-learn* package[80] v0.23.2 in Python 3.8.3 with default parameters.

To exclude the potential effect of spatial autocorrelation inherent to the transcriptomic data and the hub localization, we repeated the above described XGBoost and SVM classifiers training and testing procedures using surrogate hub identifications with the spatial autocorrelations being corrected using a generative model[81]. As shown in Supplementary Fig. 8a, we first constructed a surrogate $Z$ value map with the spatial autocorrelation being corrected using a generative model[81] based on the unthresholded $Z$ value map corresponding to the hub identification map in Fig. 1c. Then, for the 1158 AHBA brain samples within our gray matter mask, we assigned the 382 samples with the highest surrogate $Z$ values as hub samples and the 776 samples with the lowest surrogate $Z$ values as non-hub samples. For the XGBoost classifier, we trained the classifier with 300 randomly selected surrogate hub samples and 300 randomly selected surrogate non-hub samples using 10,027 genes' transcriptomic data from the preprocessed AHBA

dataset and tested it with the remaining 82 surrogate hub samples and 476 surrogate non-hub samples. We performed a 30-fold cross-validation procedure to identify the optimal number of model training iterations. We implemented XGBoost using the *XGBoost* package[33] v1.2.0.1 in R 4.0.2 with following parameters: $nrounds = 1,500$, $early\_stopping\_rounds = 50$, $eta = 0.05$, $objective = $"binary:logistic". For the SVM classifier, we built a supervised SVM classifier through a 382-fold cross-validation procedure to distinguish all 382 surrogate hub samples from 382 randomly setected surrogate non-hub samples using the transcriptomic data of the top 150 genes listed in Supplementary Data 1. We implemented SVM using the *svm* function from the *scikit-learn* package[80] v0.23.2 in Python 3.8.3 with default parameters. Finally, we repeated the surrogate hub identification generating, XGBoost classifier training and teseting, and SVM classifier training procedures 1000 times.

The top 150 key genes (Supplementary Data 1) mostly contributed to the classification results were submitted to GO enrichment analyses using GOrilla[35] and DAVID[36,37] v6.8. We conducted two GO enrichment analyses using GOrilla. The first analysis used the 150 key genes as the target list and all 10,027 genes as the background list. The second analysis used the ranked 10,027 genes according to their contributions to the XGBoost classifier. Of note, we performed GO enrichment analyses for all three ontology categories: biological process, molecular function, and cellular component. However, only analysis for biological process yielded significant GO terms. We repeated GO enrichment analysis for biological process using DAVID with the 150 key genes as the target list and all 10,027 genes as the background list. In addition, we also performed GO enrichment analysis for disease association using DAVID with the 150 key genes as the target list and all 10,027 genes as the background list.

Based on GO enrichment analysis results, we tested transcription level differences for gene sets involved in key neurodevelopmental processes[38] (Supplementary Data 6) and main neuronal metabolic pathways[39] (oxidative phosphorylation[40] and aerobic glycolysis[41], Supplementary Data 7) between connectome hubs and non-hubs through one-sided Wilcoxon rank-sum test. In line with prior studies[38,41], we used the first principal component of each gene set's transcription level to plot and to perform the statistical analysis (Fig. 6a). For illustration purposes, we normalized the first principal component of each gene set's transcription level respect to its minimum and maximum values across all brain samples to the range 0–1.

To explore developmental details, we inspected the developmental trajectory of transcription level of the above gene sets (Supplementary Data 6 and 7) in hub and non-hub regions respectively using the BrainSpan Atlas[42]. The normalized BrainSpan Atlas was generated using 524 brain samples from 42 donors aged from eight post-conceptional weeks to 40 postnatal years, including 52,376 genes' transcriptomic data from 11 neocortical areas and five additional regions of the human brain. The brain regions used in our analysis are listed in Supplementary Data 10. We plotted the developmental trajectory using locally weighted regression by smoothing the first principal component of each gene set's transcription level against log2[post-conceptional days] as in a prior study[38] (Fig. 6b). For most developmental periods, there are only no more than five hub brain samples and 10 non-hub brain samples at a specific age (Supplementary Fig. 11). Such small simple size makes it practically impossible to determine the statistical significance level of difference in transcription level between hub and non-hub regions at a specific age. We compared the magnitude of differences in developmental trajectory between hub and non-hub regions to the median absolute deviation of transcription level across brain regions at a specific age (Fig. 6c). The magnitude of differences in developmental trajectory exceeding the median absolute deviation indicates a trend of greater difference in transcription level between hub and non-hub regions than expected at a specific age. Of note, considering apparent transcriptomic differences compared to the neocortex[38], we excluded the striatum, mediodorsal nucleus of the thalamus, and cerebellar cortex in the developmental trajectory analysis but not the amygdala and hippocampus whose developmental trajectories of transcription level are more similar to those of the neocortex than to those of other subcortical structures[38]. Analysis using only neocortical regions revealed similar results (Supplementary Fig. 9).

**Contextualizing functional connectome hubs' transcriptomic pattern using established neuroimaging patterns**. We assessed the neural relevance of the above identified transcriptomic pattern underlying functional connectome hubs by contextualizing it relative to established neuroimaging patterns of neurotransmitter[46], cortical fiber length[47], brain metabolism[48], and cortical thickness atrophy in neuropsychiatric disorders[50].

The JuSpace toolbox[46] provided 15 neurotransmitter receptor and transporter density maps in the MNI volume space. For each of the 15 density maps, we tested difference in density between hub and non-hub voxels through one-sided Wilcoxon rank-sum test (Fig. 7a). For illustration purposes, we normalized the density value respect to its median and median absolute deviation across voxels.

The cortical fiber length profiling dataset[47] provided fiber number data across different length bins in a standard brain surface space. We resampled the identified hub distribution mask in Fig. 1c from the MNI volume space to the standard brain surface space provided by the dataset[47] and tested difference in fiber number between hub and non-hub vertices for each length bin through one-sided Wilcoxon

rank-sum test (Fig. 7b). For illustration purposes, we normalized the fiber number value respect to its mean and standard deviation across voxels.

The brain metabolism dataset provided by the positron emission tomography study[48] was assigned to 82 Brodmann areas in a standard brain surface space and seven subcortical structures in the MNI volume space. We first resampled the identified hub distribution mask in Fig. 1c from the MNI volume space to the standard brain surface space provided by the dataset[48] and identified Brodmann areas with more than 50% vertices within the hub distribution mask as hub regions. Then, we identified subcortical structures with more than 50% voxels within the hub distribution mask as hub regions. After that, we examined differences between hub and non-hub regions in metabolic measurements of blood supply (the cerebral blood flow), oxidative phosphorylation (the cerebral metabolic rate for oxygen), and aerobic glycolysis (the glycolytic index) through one-sided Wilcoxon rank-sum test (Fig. 7c).

The Cohen's $d$ value of cortical thickness atrophy in neuropsychiatric disorders was assigned to 68 cortical areas in a standard brain surface space[50]. We first resampled the unthresholded Cohen's $d$ map of connectome hub in Fig. 1c from the MNI volume space to the standard brain surface space provided by the dataset[50] and computed the Cohen's $d$ value for each of the 68 cortical areas by averaging Cohen's $d$ value across vertices within each cortical area. Then, we computed Pearson's correlation coefficient between the Cohen's $d$ value of connectome hub and the Cohen's $d$ value of cortical thickness atrophy across 68 cortical areas for each of the eight disorders (Fig. 7d). To reduce the potential effects of development on our results, we used cortical thickness atrophy data from adults for the attention deficit hyperactivity disorder, bipolar disorder, major depressive disorder, and obsessive-compulsive disorder.

**Statistics and reproducibility**. We performed statistical analyses using MATLAB R2013a. The statistical significances of brain clusters in Figs. 1c and 3 and Supplementary Figs. 2c, 3c, 5, and 10a were determined by comparing the observed $Z$ values with their corresponding null distributions constructed by above mentioned 10,000 one-sided nonparametric permutation tests[28]. To determine the statistical significances of one-sided Wilcoxon rank-sum tests in Figs. 6a, 7a–c and Supplementary Fig. 10e, we constructed 1000 surrogate hub identification maps with the spatial autocorrelations being corrected using a generative model[81] and repeated calculating rank-sum statistics using these surrogate hub identification maps to construct a null distribution. Then, $p$ values of these rank-sum statistics were determined by comparing the observed values with their corresponding null distributions and were Bonferroni-corrected. Surrogate hub identification maps for Figs. 6a and 7a–c were constructed based on the hub identification map in Fig. 1c. Surrogate hub identification maps for Supplementary Fig. 10e were constructed based on the hub identification map in Supplementary Fig. 10a. To determine the statistical significances of Pearson's correlation coefficients in Fig. 7d, we constructed 1000 surrogate maps of the unthresholded Cohen's $d$ map in Fig. 1c with the spatial autocorrelations being corrected using a generative model[81] and repeated calculating Pearson's correlation coefficients using these surrogate Cohen's $d$ maps to construct a null distribution. Then, $p$ values of these Pearson's correlation coefficients were determined by comparing the observed values with their corresponding null distributions.

**Reporting summary**. Further information on research design is available in the Nature Research Reporting Summary linked to this article.

## Data availability

The MRI data of the first 60 cohorts listed in Supplementary Data 8 are available at the International Neuroimaging Data-sharing Initiative (http://fcon_1000.projects.nitrc.org), Brain Genomics Superstruct Project[82] (https://doi.org/10.7910/DVN/25833), Human Connectome Project (https://www.humanconnectome.org), MPI-Leipzig Mind-Brain-Body Project (https://openneuro.org/datasets/ds000221), and Age-ility Project (https://www.nitrc.org/projects/age-ility). The MRI data of the PKU cohort are under active use by the reporting laboratory and will be available from the corresponding author upon reasonable request. The preprocessed AHBA dataset is available at https://doi.org/10.6084/m9.figshare.6852911. The normalized BrainSpan Atlas dataset is available at http://brainspan.org/static/download.html. The neurotransmitter receptor and transporter density maps provided by the JuSpace toolbox[46] are available at https://github.com/juryxy/JuSpace. The fiber length profiling dataset[47] is available at https://balsa.wustl.edu/study/1K3l. The cortical thickness atrophy dataset provided by the ENIGMA Toolbox[50] is available at https://github.com/MICA-MNI/ENIGMA. Numerical source data to reproduce all figure panels is available at https://doi.org/10.6084/m9.figshare.21194128.

## Code availability

The code to reproduce the results and visualizations of this manuscript is available on Zenodo[83]. Software packages used in this manuscript include MATLAB R2013a (https://www.mathworks.com/products/matlab.html), SPM12 v6470 (https://www.fil.ion.ucl.ac.uk/spm/software/spm12), GRETNA[72] v2.0.0 (https://www.nitrc.org/projects/gretna),

Connectome Workbench v1.4.2 (https://www.humanconnectome.org/software/connectome-workbench), cifti-matlab v2 (https://github.com/Washington-University/cifti-matlab), R 4.0.2 (https://www.r-project.org), XGBoost package[33] v1.2.0.1 (https://cran.r-project.org/web/packages/xgboost), Python 3.8.3 (https://www.python.org), and scikit-learn package[80] v0.23.2 (https://scikit-learn.org). Online analysis tools used in this manuscript include GOrilla[35] (http://cbl-gorilla.cs.technion.ac.il) and DAVID[36,37] v6.8 (https://david.ncifcrf.gov).

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

## Acknowledgements

We thank Drs. Huali Wang and Xiaodan Chen for data acquisition of the PKU cohort and Drs. Qiushi Wang and Nan Zhang for valuable advice on MRI data quality controls. This work was supported by the National Natural Science Foundation of China [82021004, 31830034, and 81620108016 to Y.H., 82071998 and 81671767 to M.X., 81971690 to X.L., 81801783 to T.Z.], Changjiang Scholar Professorship Award [T2015027 to Y.H.], National Key Research and Development Project [2018YFA0701402 to Y.H.], Beijing Nova Program [Z191100001119023 to M.X.], and Fundamental Research Funds for the Central Universities [2020NTST29 to M.X.].

## Author contributions

Conceptualization: Z.X., Y.H.; Methodology: Z.X., Y.H., M.X., X.W., X.L., T.Z.; Investigation: Z.X.; Visualization: Z.X.; Supervision: Y.H.; Writing—original draft: Z.X., Y.H.; Writing—review & editing: Y.H., Z.X., M.X., X.L., T.Z., X.W.

## Competing interests

The authors declare no competing interests.
