## [Peer Review File · Communications Biology]

Reviewers' comments:

Reviewer #1 (Remarks to the Author):

The paper by Xu et al examines the consistency and reproducibility of functional hubs, as well as relates these to transcriptomics. Using a large data set of multiple cohorts, they identify consistent hubs and show these hubs have a distinct transcriptomic signature. The issue of hub locations differing across functional studies is a problem which has not received extensive examination, despite being an issue of importance. I believe the work is likely to be of interest but have a number of suggestions which should be taken into consideration to help improve the manuscript.

Major issues:

1. Hubs are often defined based on functional activity within a region of a parcellation. This parcellation is often defined a priori or as the result of something like ICA. Connectivity between these parcels is then assessed and then hubs are examined. The approach employed by the authors is not especially common and needs some justification as to why it was used over these other approaches. These different methods of defining a region will cause variation in hub spatial location. Thus far I do not see any acknowledgement in the manuscript of these different ways of defining regions beyond a quick mention of different "analysis strategies". So for me, the issue is twofold: firstly there needs to be justification for why this method of network construction (which is outside the norm) was used, followed by a discussion of how differences in network construction can lead to differences. You may even want to suggest (or think about for future work) seeing what results are obtained when other network construction approaches are used?

2. One issue that readers may more broadly take is that defining hubs by connection strength in functional networks is problematic. Due to the transitivity of the correlation coefficient (i.e., if a is correlated with b, and b is correlated with c, then a is correlated with c) it difficult for a high degree node to show correlated activity with a node in a different module (see Power et al., 2013 for a discussion of this issue). Due to this, many papers in the current literature use the participation coefficient issue. This point should be discussed.

3. The authors argue that because they use different machine learning models and different enrichment techniques with two different transcriptomic datasets that the results found are unlikely to be false-positives. However none of these approaches appeared to have accounted for the spatial autocorrelation inherent to transcriptomic data (see Fulcher et al., 2021). While many different approaches may show the same result, if they all have the same fundamental flaw then it brings the result of all of those into question. Thus the spatial autocorrelation of transcriptomic must be accounted for in order for these results to be considered up to par with the current state of the literature, especially given the transcriptomic result receive an extensive discussion. The reference I cited above provides tools to do this.

4. The transcriptomic trajectory results appear to have only been determined by visual inspection correct? These differences don't appear to be especially pronounced as claimed, but are instead rather subtle. To establish the validity of any such findings, you could randomly assign regions to be hub or non-hub and compute the trajectories for these randomised regions to build up a null distribution. Alternatively (and probably a better idea as it will preserve spatial information) is to apply a spin permutation to the regions and measure the trajectories of these permutations. This would do better than a visual inspection. regardless, if the current results do hold, I would temper the language as the differences are not that large.

Minor issues:

5. This analysis appears to have been performed in voxel space. It would be good to mention that performing an analysis in surface space (like the Human Connectome Project does) may achieve different results.

6. It would be useful to add a quick definition as to what a transcriptomic trajectory is.

7. It is not clear what the parallel interdigitated subnetworks hypothesis is (page 9, line 295), would be good to give a primer.
8. References 34, the authors name is incorrectly formatted
9. It seems the fibre distances were in a surface projection, and not a voxel projection? How was the voxel data converted to use on the surface (or vice versa)?
10. Page 6, line 169 "The contributions of the top 300 mostly contributed key genes were consistent between the first 500 repetitions" wording is a little clunky, unclear what "mostly contributed key genes" means
11. Page 7, line 239. It is specified that hubs have more short, medium and long range fibres but no guidance is given here as to how these categories are defined and this is the first mention of them. I think you can cut "That is, hub regions have more short, medium, and long fibers, whereas non-hub regions have more very short (< 40 mm) fibers," and just go straight to "suggesting a more...."
12. Page 8, line 275 "cohorts" instead of "cohort"
13. Page 11, line 404 It should read "in line with a previous neuroimaging meta-analysis study" (the "a" is missing originally)
14. Page 12, line 429 "Then, We". The "we" should not be capitalised
15. Page 13, line 451 "samples to range from 0 to 1" should be "samples to the range 0 to 1"

References

Power JD, Schlaggar BL, Lessov-Schlaggar CN, Petersen SE. Evidence for hubs in human functional brain networks. *Neuron*. 2013 Aug 21;79(4):798-813. doi: 10.1016/j.neuron.2013.07.035. PMID: 23972601; PMCID: PMC3838673.

Fulcher, B.D., Arnatkeviciute, A. & Fornito, A. Overcoming false-positive gene-category enrichment in the analysis of spatially resolved transcriptomic brain atlas data. *Nat Commun* 12, 2669 (2021). <https://doi.org/10.1038/s41467-021-22862-1>

Reviewer #2 (Remarks to the Author):

The authors use a worldwide harmonized meta-connectomic analysis of 5,212 healthy young adults across 61 cohorts, providing consistent and reproducible functional connectome hubs in the resting human brain. Using transcriptomic data from the AHBA and BrainSpan Atlas, they found that these connectome hubs have a spatiotemporally distinctive transcriptomic pattern in contrast to non-hub regions. The methods are solid and well-conducted, and the findings are important which advanced our understanding of the underlying cellular and molecular mechanism of macroscopic functional connectome hubs. Here, I have several minor comments, which I hope are useful to improve the manuscript.

1. Higher brain function requires the integration of distributed neuronal activity across large-scale brain networks. Recent network analyses that have demonstrated that the basal ganglia and thalamus belong to an ensemble of highly interconnected network hubs, which form part of a core circuit that supports large-scale integration of functionally diverse neural signals. However, the authors did not find any subcortical brain regions belonging to connectome hubs. Please explain.
2. Most (27.5%) connectome hubs belong to the DMN network. Does this result only reflect the characteristics of the brain network in a resting state?
3. Whether the connectome hubs are consistent by using dynamic functional connections, and is there a difference in temporal variability between hub regions and non-hub regions?
4. Could transcriptomic data distinguish three clusters of connectome hubs?
5. Is there a difference in cell-type density between hub regions and non-hub regions? Cell-class

density proxy maps can be generated from bulk-tissue AHBA expression data using information from single-cell gene expression studies.

6. These brain hubs have a spatiotemporally distinctive transcriptomic pattern dominated by genes with the highest enrichment for the neuropeptide signalling pathway. These results will be more reliable if supported by PET imaging, like dopamine, norepinephrine, serotonin, acetylcholine, glutamate, GABA, histamine, cannabinoid, and opioid (JuSpace, 2020).

7. "Information flow along the primary visual, visual association, and higher-level sensorimotor cortices is undertaken by the four occipital hubs (Cluster II) left VMV1, right V4, and bilateral V3A that are all densely connected with the VIS and portions of the SMN, DAN, and VAN." This sentence needs the support of references.

8. There needs to be a discussion of the relationship between FCS maps and ENIGMA-derived patterns of brain atrophy across neurological, psychiatric, and neurodevelopmental disorders and what will be useful in clinical practice and informing the design of future studies.

Manuscript COMMSBIO-22-1399-T

Title:

Mapping consistent, reproducible, and transcriptionally relevant functional connectome hubs of the human brain

Legend:

Comment from reviewer.

Response to reviewer's comment.

Text appeared in the previous manuscript version.

New text appears in the revised manuscript version.

Text removed from the revised manuscript but included in the response for convenience.

Page and line numbers refer to the revised manuscript.

Reviewer #1:

Remarks to the Author:

The paper by Xu et al examines the consistency and reproducibility of functional hubs, as well as relates these to transcriptomics. Using a large data set of multiple cohorts, they identify consistent hubs and show these hubs have a distinct transcriptomic signature. The issue of hub locations differing across functional studies is a problem which has not received extensive examination, despite being an issue of importance. I believe the work is likely to be of interest but have a number of suggestions which should be taken into consideration to help improve the manuscript.

Response: We thank the reviewer for the positive appraisal and the valuable comments to further improve the quality of our manuscript. We have carefully considered all points and followed the suggestions to revise the Main Manuscript and the Supplementary Information to improve clarity of our interpretation of the results as described in more detail below.

1.1. Hubs are often defined based on functional activity within a region of a parcellation. This parcellation is often defined a priori or as the result of something like ICA. Connectivity between these parcels is then assessed and then hubs are examined. The approach employed by the authors is not especially common and needs some justification as to why it was used over these other approaches. These different methods of defining a region will cause variation in hub spatial location. Thus far I do not see any acknowledgement in the manuscript of these different ways of defining regions beyond a quick mention of different “analysis strategies”. So for me, the issue is twofold: firstly there needs to be justification for why this method of network construction (which is outside the norm) was used, followed by a discussion of how differences in network construction can lead to differences. You may even want to suggest (or think about for future work) seeing what results are obtained when other network construction approaches are used?

Response: We agree with the reviewer that the construction of brain network is often based on a priori parcellation. We adopted a voxel-based analysis framework based on the following three considerations. First, the choice of parcellation scheme has an effect on the distribution shape of functional connectivity strength (Figure 2 in ref¹) and the location of highly connected brain regions (Figure 5 in ref²). A voxel-based analysis made it feasible to directly compare our results with the extant voxel-based reports³⁻¹⁰ and to further analyze potential causes for inconsistent and less reproducible hub localizations among these extant reports. Second, a voxel-based connectome analysis can increase the sensitivity of identifying spatially focal hubs¹¹, such as candidate hubs with no more than 10 voxels reported in prior studies^{4, 8}. However, almost all extant fMRI- and multimodal MRI-based parcellations divided about hundreds of thousands of voxels into 200 to 500 parcels¹² where each consists of hundreds of voxels. These candidate voxel-sized hubs may be undetectable in a connectome analysis using extant fMRI- and multimodal MRI-based parcellations. Third, parcellations defined a priori “generally capture the main aspects of organization evident across individuals, whereas the size, shape and position of areas and networks can vary substantially between individuals”¹². A connectome analysis using parcellations defined a priori may underestimate the inter-individual variance within each cohort, which will substantially bias the random-effects meta-analysis across cohorts.

We also agree with the reviewer that the differences in brain network construction approaches may cause variation in hub localizations. This issue will be an important topic in our future work.

We have revised the Main Manuscript to make these two issues clear.

Main Manuscript, Discussion, page 11, lines 404-408:

[...] Second, we conducted a voxel-based connectome analysis in order to directly compare our results with the extant voxel-based reports^{7, 8, 14-19} and increase the sensitivity of identifying spatially focal (e.g., voxel-sized) hubs⁷⁰. The effects of parcellation-based⁷⁰ and surface-based⁷¹ analysis on hub localizations should be resolved in future studies. Third, the AHBA dataset [...]

1.2. One issue that readers may more broadly take is that defining hubs by connection strength in functional networks is problematic. Due to the transitivity of the correlation coefficient (i.e., if a is correlated with b, and b is correlated with c, then a is correlated with c) it difficult for a high degree node to show correlated activity with a node in a different module (see Power et al., 2013 for a discussion of this issue). Due to this, many papers in the current literature use the participation coefficient issue. This point should be discussed.

Response: We have noted the argument in Power et al., 2013 that identifying hub regions based on connection strength may highlight only members of large brain modules due to the difficulty of showing correlations with nodes outside of their modules¹³. But we observed all hub peaks possessing large amounts of inter-module connections (Figure 3 and Figure S7). Our validation analysis demonstrated no significant correlation between a voxel's FCS and the size of the brain module to which it belongs (Spearman's $\rho \leq 0.231$, Bonferroni-corrected $ps \geq 0.285$, Figure S4), suggesting that functional connectome hubs identified using FCS in the present study were not driven by the size of the brain module to which they belong. We speculated that the conclusion in Power et al., 2013 may be driven by unreasonable connection threshold (Pearson's r : 0.20-0.37)¹³ because we observed all hub peaks possessing significant connections with Pearson's r less than 0.2 for both intra- and inter-module connections (Figure S7). These validation analyses have been mentioned in the Main Manuscript and Supplementary Information, which are listed below.

We agree with the reviewer that the participation coefficient is a useful hub identification measurement. We employed the functional connectivity strength to define hub regions because of its wide usage in prior studies^{3-10, 14} and its high spatial similarity compared with other measures (Figure 2B in ref¹⁵). We have revised the Supplementary Information to make this issue clear.

Main Manuscript, Results, page 5, lines 133-135:

Validation analysis demonstrated that the above results [...] were not driven by the size of the brain network to which they belong³¹ (Figure S4) [...]

Supplementary Information, Supplementary Text II, page 7, lines 209-215:

A prior study argued that identifying hub regions based on FCS highlights only members of large brain networks rather than brain regions playing crucial roles in global brain communication²³. However, no significant correlation between a voxel's FCS and the size of the brain network to which it belongs could be identified in the present study (Figure S4). The conclusion in the study²³ may be driven by unreasonable connection threshold (Pearson's r : 0.20-0.37) because we observed all hub peaks possessing significant connections with Pearson's r less than 0.2 (Figure S7).

Supplementary Information, Supplementary Text II, page 7, lines 216-220:

We employed the functional connectivity strength to identify hub regions because of its wide usage in prior studies^{15-20, 25, 26, 31} and its high spatial similarity compared with other measures³². There are other useful hub identification measures, like the participation coefficient that may report more candidate hubs in subcortical structures because of their involvements in diverse functional domains, such as the basal ganglia³³ and the thalamus³⁴.

1.3. The authors argue that because they use different machine learning models and different enrichment techniques with two different transcriptomic datasets that the results found are unlikely to be false-positives. However none of these approaches appeared to have accounted for the spatial autocorrelation inherent to transcriptomic data (see Fulcher et al., 2021). While many different approaches may show the same result, if they all have the same fundamental flaw then it brings the result of all of those into question. Thus the spatial autocorrelation of transcriptomic must be accounted for in order for these results to be considered up to par with the current state of the literature, especially given the transcriptomic result receive an extensive discussion. The reference I cited above provides tools to do this.

Response: We agree with the reviewer that the spatial autocorrelation inherent to the hub localization and the transcriptomic data may cause different analysis approaches showing the same result. We have constructed 1,000 surrogate hub identification maps with the spatial autocorrelations being corrected using the same generative model¹⁶ as in Fulcher et al., 2021¹⁷ and repeated training supervised machine learning classifiers based on XGBoost¹⁸ and support vector machine to distinguish surrogate hubs from surrogate non-hubs using transcriptomic data from the AHBA dataset. These classifiers trained using surrogate hub identification maps performed no better than the chance level (Figure S8), confirming that the performance of the XGBoost and SVM classifiers in our main analyses was not driven by the effects of spatial autocorrelation. We have revised the Main Manuscript and Supplementary Information according to these additional validation analyses.

Main Manuscript, Results, page 6, lines 185-189:

Validation analyses showed that the XGBoost and SVM classifiers trained using surrogate hub identification maps with the spatial autocorrelations being corrected performed no better than the chance level (Figure S8), confirming that the performance of the XGBoost and SVM classifiers was not driven by the effects of spatial autocorrelation inherent to the hub localization and the transcriptomic data. [...]

Supplementary Information, Supplementary Text I, page 5, lines 124-141:

The effects of spatial autocorrelation. To exclude the potential effect of spatial autocorrelation inherent to the transcriptomic data and the hub localization, we repeated the above described XGBoost and SVM classifiers training and testing procedures using surrogate hub identifications with the spatial autocorrelations being corrected using a generative model¹¹. As shown in Figure S8A, we firstly constructed a surrogate Z value map based on the unthreshold Z value map corresponding to the hub identification map in Fig 2B with the spatial autocorrelation being corrected using a generative model¹¹. Then, for the 1,158 AHBA brain samples within our gray matter mask, we assigned the 382 samples with the highest surrogate Z values as hub samples and the 776 samples with the lowest surrogate Z values as non-hub samples. For the XGBoost classifier, we built a supervised XGBoost classifier through a 30-fold cross-validation procedure to distinguish 300 randomly selected hub samples from 300 randomly selected non-hub samples using 10,027 genes' transcriptomic data from the preprocessed AHBA dataset and tested the XGBoost classifier with the remaining 82 hub samples and 476 non-hub samples. For the SVM classifier, we built a supervised SVM classifier through a 382-fold cross-validation procedure to distinguish all 382 hub samples from 382 randomly

selected non-hub samples using the transcriptomic data of the top 150 genes listed in Table S1. Finally, we repeated the surrogate hub identification generating, XGBoost classifier training and testing, and SVM classifier training procedures 1,000 times. We implemented the XGBoost and SVM as described above.

Supplementary Information, Supplementary Figures, page 19, lines 411-416:

Figure S8. Transcriptomic data cannot distinguish surrogate hubs from surrogate non-hubs. A Schematic diagram of generating surrogate hub identification and using XGBoost and SVM classifiers to distinguish surrogate hub samples from surrogate non-hub samples. B Performance of the XGBoost classifier. Each dot represents one repetition in A. The horizontal gray dashed line represents the chance level accuracy rate (50%). C Performance of the SVM classifier. Each dot represents one repetition in A. The horizontal gray dashed line represents the chance level accuracy rate (50%).

1.4. The transcriptomic trajectory results appear to have only been determined by visual inspection correct? These differences don't appear to be especially pronounced as claimed, but are instead rather subtle. To establish the validity of any such findings, you could randomly assign regions to be hub or non-hub and compute the trajectories for these randomised regions to build up a null distribution. Alternatively (and probably a better idea as it will preserve spatial information) is to apply a spin permutation to the regions and measure the trajectories of these permutations. This would do better than a visual inspection. regardless, if the current results do hold, I would temper the language as the differences are not that large.

Response: We agree with the reviewer's concern about the validity of differences in transcription level between hub and non-hub regions during development. We have tried to find a suitable statistical model to examine the validity of differences in transcription level at the beginning of our study. But for most developmental periods, there are only no more than five hub brain samples and 10 non-hub brain samples at a specific age (Figure S11). Such small sample size makes it practically impossible to determine the statistical significance level of difference in transcription level between hub and non-hub regions at a specific age. Even for a non-parameter permutation test, such small sample size is inadequate to perform an effective randomization or spin test. Thus, we adopted a compromise approach by comparing the magnitude of differences in developmental trajectory between hub and non-hub regions to the median absolute deviation of transcription level across brain regions at a specific age (Figure 5C). The magnitude of differences in developmental trajectory exceeding the median absolute deviation indicates a trend of greater difference in transcription level between hub and non-hub regions than expected at a specific age. We have revised the Main Manuscript and Supplementary Information according to these additional analyses.

Main Manuscript, Results, page 7, lines 221-235:

[...] To explore their developmental evolutions, we inspected the developmental trajectory of transcription level in hub and non-hub regions respectively using the BrainSpan Atlas⁴². We observed diverging developmental trajectories of transcription level between hub and non-hub regions in these key neurodevelopmental processes and main neuronal metabolic pathways (Figure 5B and Figure S9A). The magnitude of differences in developmental trajectory between hub and non-hub regions continuously exceeds the median absolute deviation of transcription level across brain regions during some periods (Figure 5C and Figure S9B), suggesting a trend of greater difference than expected. Specifically, hub regions have higher transcription levels for neuron migration during the late-fetal period, higher transcription levels for dendrite and synapse development from the late-childhood to mid-adolescence period, and lower transcription levels for axon development and myelination from the mid-childhood to late-adolescence period than non-hub regions. ~~we inspected regional transcriptomic trajectory differences between hub and non-hub regions in these key neurodevelopmental processes using the BrainSpan Atlas⁴². We observed pronounced diverging transcriptomic trajectories between hub and non-hub regions in these key neurodevelopmental processes and main neuronal metabolic pathways for genes associated with neuron migration, dendrite, synapse, axon development, and myelination but not for neuron differentiation (Fig 5B and Fig S8). For neuron migration, the transcription level in hub regions is higher than that in non-hub regions during from the latemid fetal period and to early infancy. For~~

~~dendrite, synapse, axon development, and myelination, transcriptomic trajectories of hub regions apparently diverge from those of non-hubs since the beginning of childhood and diminishes at the end of adolescence, during which hub regions have higher transcription levels for dendrite and synapse development but lower transcription levels for axon development and myelination.~~ These results are in agreement with the observation of primary somatosensory, auditory, and visual (V1/V2) cortices with lower synapse density but higher myelination than the prefrontal area^{43, 44}. Moreover, hub regions have higher transcription levels than non-hub regions for aerobic glycolysis since the early childhood period ~~and for oxidative phosphorylation during childhood and adolescence.~~ [...]

Main Manuscript, Methods, page 14, lines 496-513:

To explore developmental details, we inspected the developmental trajectory of transcription level of the above gene sets in hub and non-hub regions respectively using the BrainSpan Atlas⁴². ~~we inspected transcriptomic trajectory differences between connectome hubs and non-hubs in the above gene sets using the BrainSpan Atlas⁴². In line with prior studies^{38, 41}, we used the first principal component of each gene set's transcription level to plot transcriptomic trajectories and visually inspected transcriptomic trajectory differences between connectome hubs and non-hubs (Fig 5B). Transcriptomic trajectories were plotted~~ We plotted the developmental trajectory using locally weighted regression by smoothing the first principal component of each gene set's transcription level against log₂[post-conceptual days] as in a prior study³⁸ (Figure 5B). For most developmental periods, there are only no more than 5 hub brain samples and 10 non-hub brain samples at a specific age (Figure S10). Such small sample size makes it practically impossible to determine the statistical significance level of difference in transcription level between hub and non-hub regions at a specific age. We compared the magnitude of differences in developmental trajectory between hub and non-hub regions to the median absolute deviation of transcription level across brain regions at a specific age (Figure 5C). The magnitude of differences in developmental trajectory exceeding the median absolute deviation indicates a trend of greater difference in transcription level between hub and non-hub regions than expected at a specific age. Of note, considering apparent transcriptomic differences compared to the neocortex³⁸, we excluded the striatum, mediodorsal nucleus of the thalamus, and cerebellar cortex in the developmental ~~transcriptomic~~ trajectory analysis but not the amygdala and hippocampus whose developmental trajectories of transcription level ~~transcriptomic trajectories~~ are more similar to those of the neocortex than to those of other subcortical structures³⁸. Analysis using only neocortical regions revealed similar ~~almost unchanged~~ results (Figure S9).

Main Manuscript, Figures, pages 30-31, lines 893-904:

Figure 5. Connectome hubs have a spatiotemporally distinctive transcriptomic pattern.

A Transcription level differences between hub samples (n=382) and non-hub samples (n=776) for genes associated with key neurodevelopmental processes³⁸ and main neuronal metabolic pathways³⁹. Boxplot edges, gray lines, and whiskers and dots depict the 25th and 75th percentiles, median, and extreme nonoutlier and outlier values, respectively. Significance of one-sided Wilcoxon rank-sum tests were determined by 1,000 permutation tests and were labeled with Bonferroni-corrected p values. **B** *Developmental trajectory of transcription level in ~~Transcriptomic trajectory differences between~~ hub and non-hub regions for genes involved in key neurodevelopmental processes³⁸ and main neuronal metabolic pathways³⁹.* **C** *Differences in the developmental trajectory of transcription level between hub regions and that in non-hub regions shown in B.* MAD, the median absolute deviation of transcription level across brain regions. w, post-conceptional week; y, postnatal year; a.u., arbitrary unit.

Supplementary Information, Supplementary Figures, page 20, lines 417-422:

Supplementary Information, Supplementary Figures, page 22, lines 439-440:

Figure S11. Age distribution of brain samples from the BrainSpan Atlas dataset. *w*, post-conceptional week; *y*, postnatal year.

1.5. This analysis appears to have been performed in voxel space. It would be good to mention that performing an analysis in surface space (like the Human Connectome Project does) may achieve different results.

Response: We agree with the reviewer that the voxel space-based analysis may achieve subtle differences compared with the surface-based analysis. We have revised the Main Manuscript to make this issue clear.

Main Manuscript, Discussion, page 11, lines 404-408:

[...] Second, we conducted a voxel-based connectome analysis in order to directly compare our results with the extant voxel-based reports^{7, 8, 14-19} and increase the sensitivity of identifying spatially focal (e.g., voxel-sized) hubs⁶⁷. The effects of parcellation-based⁶⁷ and surface-based⁶⁸ analysis on hub localizations should be resolved in future studies. Third, the AHBA dataset [...]

1.6. It would be useful to add a quick definition as to what a transcriptomic trajectory is.

Response: We have replaced “transcriptomic trajectory” with a more straightforward phrase “developmental trajectory of transcription level”.

Main Manuscript, Results, page 7, lines 221-222:

[...] To explore their developmental evolutions, we inspected the developmental trajectory of transcription level in hub and non-hub regions respectively using the BrainSpan Atlas⁴². ~~we inspected regional transcriptomic trajectory differences between hub and non-hub regions in these key neurodevelopmental processes using the BrainSpan Atlas⁴².~~ [...]

1.7. It is not clear what the parallel interdigitated subnetworks hypothesis is (page 9, line 295), would be good to give a primer.

Response: We have added a description to make it clear.

Main Manuscript, Discussion, page 10, lines 330-333:

[...] This can be supported by the recent finding of a control-default connector located in the posterior middle frontal gyrus³² and may also be a case of the hypothesis of parallel interdigitated subnetworks⁵⁷ where the posterior middle frontal gyrus is connected with a subnetwork of the DMN and some regions of the FPN. [...]

1.8. References 34, the authors name is incorrectly formatted

Response: This error has been fixed in the revision.

Main Manuscript, References, page 17, line 709:

34. *Arnatkevičiute*~~Arnatkevičute~~ A, Fulcher BD, Fornito A. [...]

1.9. It seems the fibre distances were in a surface projection, and not a voxel projection? How was the voxel data converted to use on the surface (or vice versa)?

Response: We resampled the identified hub distribution mask in Figure 2B from the MNI volume space to the standard brain surface space provided by the cortical fiber length profiling dataset¹⁹. We have revised the Main Manuscript to make it clear.

Main Manuscript, Methods, page 14, lines 525-530:

The cortical fiber length profiling dataset⁴⁷ provided fiber number data across different length bins in a standard brain surface space. We resampled the identified hub distribution mask in Figure 2B from the MNI volume space to the standard brain surface space provided by the dataset⁴⁷ and tested difference in fiber number between hub and non-hub vertices for each length bin through one-sided Wilcoxon rank-sum test (Figure 6B). For illustration purposes, we normalized the fiber number value respect to its mean and standard deviation across voxels.

1.10. Page 6, line 169 “The contributions of the top 300 mostly contributed key genes were consistent between the first 500 repetitions” wording is a little clunky, unclear what “mostly contributed key genes” means

Response: We have rephrased the sentence to make it clear.

Main Manuscript, Results, page 6, lines 170-172:

[...] The contributions of the top 300 ~~mostly contributed key~~ genes with the greatest contributions to the XGBoost classifier were consistent between the first 500 repetitions and the second 500 repetitions [...]

1.11. Page 7, line 239. It is specified that hubs have more short, medium and long range fibres but no guidance is given here as to how these categories are defined and this is the first mention of them. I think you can cut “That is, hub regions have more short, medium, and long fibers, whereas non-hub regions have more very short (< 40 mm) fibers,” and just go straight to “suggesting a more....”

Response: We have deleted redundant content in the revision.

Main Manuscript, Results, page 8, lines 256-260:

[...] Using a fiber length profiling dataset⁴⁷, we observed that hub regions possess more fibers with a length exceeding 40 mm but less fibers with a length shorter than 40 mm (one-sided Wilcoxon rank-sum tests, Bonferroni-corrected $p \leq 0.007$, Figure 6B). ~~That is, hub regions have more short, medium, and long fibers, whereas non-hub regions have more very short (< 40 mm) fibers,~~ suggesting a more intricate fiber configuration in hub regions.

1.12. Page 8, line 275 “cohorts” instead of “cohort”

Response: This error has been fixed in the revision.

Main Manuscript, Discussion, page 9, lines 309-310:

Finally, we used harmonized image processing and connectome analysis protocols across cohorts, [...]

1.13. Page 11, line 404 It should read “in line with a previous neuroimaging meta-analysis study” (the “a” is missing originally)

Response: This error has been fixed in the revision.

Main Manuscript, Methods, page 12, line 448:

In line with a previous neuroimaging meta-analysis study⁷⁴, [...]

1.14. Page 12, line 429 “Then, We”. The “we” should not be capitalized

Response: This error has been fixed in the revision.

Main Manuscript, Methods, page 13, line 473:

[...] Then, ~~W~~we compared the Fisher’s z value [...]

1.15. Page 13, line 451 “samples to range from 0 to 1” should be “samples to the range 0 to 1”

Response: This error has been fixed in the revision.

Main Manuscript, Methods, page 14, line 495:

samples to the range 0 to 1.

Reviewer #2:**Remarks to the Author:**

The authors use a worldwide harmonized meta-connectomic analysis of 5,212 healthy young adults across 61 cohorts, providing consistent and reproducible functional connectome hubs in the resting human brain. Using transcriptomic data from the AHBA and BrainSpan Atlas, they found that these connectome hubs have a spatiotemporally distinctive transcriptomic pattern in contrast to non-hub regions. The methods are solid and well-conducted, and the findings are important which advanced our understanding of the underlying cellular and molecular mechanism of macroscopic functional connectome hubs. Here, I have several minor comments, which I hope are useful to improve the manuscript.

Response: We thank the reviewer for the appreciation and the encouraging feedback to further improve the quality of our manuscript. We have followed the recommends and carefully revised the Main Manuscript and the Supplementary Information according to each point as described in more detail below.

2.1. Higher brain function requires the integration of distributed neuronal activity across large-scale brain networks. Recent network analyses that have demonstrated that the basal ganglia and thalamus belong to an ensemble of highly interconnected network hubs, which form part of a core circuit that supports large-scale integration of functionally diverse neural signals. However, the authors did not find any subcortical brain regions belonging to connectome hubs. Please explain.

Response: We agree with the reviewer that both the basal ganglia and thalamus have been reported possessing functional connections with distributed cortical regions^{20, 21}. We speculated that more complex sampling error and intercohort heterogeneity in subcortical structures caused their absence as a candidate hub in the present study. First, a prior report²² demonstrated reliable estimation of subcortical-cortical functional connections requiring more data (~100 min per subject) than conventional quantities of rsfMRI data (5–20 min per subject) adopted by prior reports^{20, 21}. In addition, individual features contribute to ~60% of the variance in subcortical-cortical functional connections²², which is higher than ~35% in cortical-cortical²³ and ~45% in cerebellar-cortical²⁴ functional connections. These two factors elevate both sampling error and intercohort heterogeneity of functional connections in subcortical structures, which is in line with our observation of higher heterogeneity among cohorts in most subcortical structures than in the cortex (Figure S1). In the random-effects meta-analysis framework, higher sampling error and intercohort heterogeneity will substantially undermine the effect size in subcortical structures, which may be the main reason of their absence as a candidate hub in the present study. These have been mentioned in the Supplementary Information, which are listed below.

Supplementary Information, Supplementary Text II, page 6, lines 173-185:

Most subcortical structures have also been argued as candidate hubs, including the thalamus^{16, 17, 25}, basal ganglia^{16, 26}, amygdala^{17, 26}, and hippocampus²⁶. Nevertheless, no subcortical structure was identified as a candidate hub in the present study. The inconsistency may be attributed to more complex sampling error and intercohort heterogeneity in subcortical structures. First, a prior report²⁷ demonstrated reliable estimation of subcortical-cortical functional connections requiring more data (~100 min per subject) than conventional quantities of rsfMRI data (5–20 min per subject) adopted by prior reports^{16, 17, 25, 26}. In addition, individual features contribute to ~60% of the variance in subcortical-cortical functional connections²⁷, which is higher than ~35% in cortical-cortical²⁸ and ~45% in cerebellar-cortical²⁹ functional connections. It precludes reliable estimation of subcortical functional connections with only dozens of subjects. These two factors complicate both sampling error and intercohort heterogeneity in subcortical structures, which is in line with our observation of higher heterogeneity among cohorts in most subcortical structures than in the cortex (Figure S1).

2.2. Most (27.5%) connectome hubs belong to the DMN network. Does this result only reflect the characteristics of the brain network in a resting state?

Response: We speculated that most connectome hubs belonging to the default-mode network is not driven by the characteristics of the brain network in a resting state but is a reflection of association cortices' vital role in supporting the organization of functional connectome.

First, we computed the voxel percentage of the eight brain networks for all 47,619 gray matter voxels (Figure R1A) and for the 15,461 hub voxels (Figure R1B). Then, we found no significant correlation between these two sets of voxel percentage (Spearman's $\rho = 0.595$, $p = 0.132$, Figure R1C). It suggests that the voxel percentage of the eight brain networks for the 15,461 hub voxels is not driven by the characteristics of the brain network in a resting state.

Finally, we conducted a fold enrichment analysis by dividing the voxel percentage of the eight brain networks for the 15,461 hub voxels by the voxel percentage of the eight brain networks for all 47,619 gray matter voxels and observed that the ventral attention, dorsal attention, default-mode, and frontoparietal networks have fold enrichment scores greater than 1 while the somatomotor, visual, limbic, and subcortical networks have fold enrichment scores smaller than 1 (Figure R1D). It suggests that connectome hubs consist of more regions of association cortices than the chance level but less regions of primary, limbic, and subcortical cortices than the chance level. Thus, we speculated that most connectome hubs belonging to the default-mode network and other association cortices is a reflection of association cortices' vital role in supporting the organization of functional connectome.

Figure R1. Connectome hubs consist of more regions of association cortices than the chance level. A and B Voxel percentage of the eight brain networks for all 47,619 gray matter voxels (A) and for the 15,461 hub voxels (B). **C** Scatter plot of the two sets of voxel percentage in A and B. Each dot represents one brain network. **D** Fold enrichment analysis of hub voxels.

2.3. Whether the connectome hubs are consistent by using dynamic functional connections, and is there a difference in temporal variability between hub regions and non-hub regions ?

Response: We agree with the reviewer that dynamic functional connections are essential for our understanding and interpretation of hub regions' vital role in supporting the connectome organization. Considerable differences in temporal variability of functional connections and modular architectures have been observed among primary, unimodal, and heteromodal regions²⁵. We speculated that there is meaningful difference in temporal variability between hub and non-hub regions. But the issue of dynamic functional connections of connectome hubs is outside of the scope of the present study to directly compare our results with the extant static functional connection reports^{3-10, 14}. This issue will be an important topic in our future work.

2.4. Could transcriptomic data distinguish three clusters of connectome hubs?

Response: We have tried to train machine learning classifier based on XGBoost¹⁸ to distinguish three clusters of connectome hubs using the AHBA dataset²⁶ at the beginning of our study (Figure R2A). But these classifiers performed no better than the chance level (Figure R2B). It may be attributed to two sources. First, the difference in transcriptome among these three clusters of connectome hubs is too subtle in bulk tissue transcriptomic data from the AHBA dataset. Second, no more than 200 brain samples for each cluster of connectome hubs may be inadequate to train a machine learning classifier.

Figure R2. Transcriptomic data cannot distinguish three clusters of connectome hubs. A Schematic diagram of using the XGBoost model to distinguish three clusters of connectome hubs. **B** Performance of the XGBoost classifier. Each dot represents one repetition in **A**.

2.5. Is there a difference in cell-type density between hub regions and non-hub regions? Cell-class density proxy maps can be generated from bulk-tissue AHBA expression data using information from single-cell gene expression studies.

Response: We thank the reviewer for this valuable suggestion. We conducted sets of cell-type specificity analyses but failed to obtain any consistent or reproducible observations.

Analysis 1: We estimated each gene's transcription level in seven types of brain cell using a single-nucleus transcriptome dataset provided by the Allen Institute for Brain Science (49,495 nuclei, <https://portal.brain-map.org/atlas-and-data/rnaseq/human-multiple-cortical-areas-smart-seq>). We identified cell-type specific genes according to their highest transcription levels across the seven types of brain cell. Both for the top 150 key genes and for all 10,027 genes, most genes were identified as neuron specific genes (Figure R3A top). We calculated fold enrichment for each type of brain cell by dividing its percent in the top 150 key genes to its percent in all 10,027 genes. Permutation tests showed no statistically significant fold enrichment (uncorrected p s > 0.075) (Figure R3A bottom). It suggests that there is no cell-type specificity for the top 150 key genes.

Analysis 2: We conducted a gene co-expression network analysis of the top 150 key genes using WGCNA (<https://horvath.genetics.ucla.edu/html/CoexpressionNetwork/Rpackages/WGCNA/>) and found 2 modules (Figure R3B top). GO enrichment analysis for both modules showed no reproducible GO term. It may be caused by inadequate target genes (50 genes in module I, 45 genes in module II). For both modules, most genes were also identified as neuron specific genes (Figure R3B bottom).

Analysis 3: For both modules identified in Analysis 2, we examined the difference in transcription level of neuron specific genes between hub and non-hub regions. Hub regions showed higher transcription level in module I but lower transcription level in module II (Figure R3C). Thus, there is no consistent evidence for higher transcription level for neuron specific genes in hub or non-hub regions. For genes out of module I and module II, the spatial similarity of their transcription levels is too low to obtain any consistent evidence for each type of brain cell.

Together, sets of cell-type specificity analyses showed no consistent or reproducible evidence for difference in cell-type density between hub and non-hub regions.

Figure R3. Cell-type specificity analyses of the top 150 key genes. **A** Top: Cell-type specific genes were identified according to their highest transcription levels across the 7 types of brain cell in a single-nucleus transcriptome dataset provided by the Allen Institute for Brain Science (49,495 nuclei, <https://portal.brain-map.org/atlas-and-data/rnaseq/human-multiple-cortical-areas-smart-seq>). Bottom: Cell-type enrichment analysis for the top 150 key genes. Uncorrected p values were estimated by 1,000 permutation tests. **B** Top: Gene co-expression network analysis of the top 150 key genes. Bottom: Cell-type genes were identified for genes in the two co-expression modules. **C** Differences in transcription level of neuron specific genes between hub and non-hub regions.

2.6. These brain hubs have a spatiotemporally distinctive transcriptomic pattern dominated by genes with the highest enrichment for the neuropeptide signalling pathway. These results will be more reliable if supported by PET imaging, like dopamine, norepinephrine, serotonin, acetylcholine, glutamate, GABA, histamine, cannabinoid, and opioid (JuSpace, 2020).

Response: We thank the reviewer for this valuable suggestion. We have added a neurotransmitter system analysis using neurotransmitter receptor and transporter maps derived from positron emission tomography and single photon emission computed tomography provided by the JuSpace toolbox²⁷.

Main Manuscript, Introduction, page 3, lines 73-77:

[...] To uncover the genetic signatures underlying these connectome hubs, we conducted machine learning approaches to distinguish connectome hubs from non-hubs using transcriptomic data from the Allen Human Brain Atlas (AHBA), explored their developmental evolutions using the BrainSpan Atlas, and assessed their neural relevance by contextualizing them relative to established neuroimaging patterns.

Main Manuscript, Results, pages 7-8, lines 240-252:

Neural contextualization of connectome hubs' transcriptomic pattern.

To assess the neural relevance of the above identified transcriptomic pattern underlying functional connectome hubs, we contextualized it relative to prior established neuroimaging maps. The identified transcriptomic pattern is dominated by genes with the highest enrichment for the neuropeptide signaling pathway. Considering that neuropeptides are a main type of indirect neurotransmitter widely distributed in the human central nervous system and their vital role in modulating direct excitatory and inhibitory transmission⁴⁵, it is rational to speculate that there are significant differences in neurotransmitter systems between hub and non-hub regions. Using neurotransmitter maps derived from positron emission tomography and single photon emission computed tomography⁴⁶, we found that hub regions have higher density of GABA_A, glutamate, mu opioid, cannabinoid, dopamine D2, and serotonin receptor and norepinephrine transporter but lower density of dopamine transporter and fluorodopa than non-hub regions (one-sided Wilcoxon rank-sum tests, Bonferroni-corrected $p_s \leq 0.015$, Figure 6A).

Main Manuscript, Discussion, page 10, lines 344-345:

[...] This is also supported by our observation of differences in neurotransmitter receptor and transporter density between hub and non-hub regions. [...]

Main Manuscript, Methods, page 14, lines 520-524:

The JuSpace toolbox⁴⁶ provided 15 neurotransmitter receptor and transporter density maps in the MNI volume space. For each of the 15 density maps, we tested difference in density between hub and non-hub voxels through one-sided Wilcoxon rank-sum test (Figure 6A). For illustration purposes, we normalized the density value respect to its median and median absolute deviation across voxels.

Main Manuscript, Figures, pages 32-33, lines 906-918:

Figure 6. Neural contextualization of connectome hubs' transcriptomic pattern. A-C

*Differences between hub (red) and non-hub (blue) regions in density of neurotransmitter receptor and transporter (A, hub voxels n=15,461, non-hub voxels n=32,158), fiber number for different fiber length bins (B, hub vertices n=25,944, non-hub vertices n=33,195), and metabolic rate for oxygen, aerobic glycolysis, and blood supply (C, hub regions n=29, non-hub regions n=60). For each violin plot, dashed gray lines depict the 25th and 75th percentiles, solid gray line depicts median value. Significance of one-sided Wilcoxon rank-sum tests was determined by 1,000 permutation tests and was labeled with Bonferroni-corrected p value. *p<0.05, **p<0.01, ***p<0.001. a.u., arbitrary unit. D Regression plot of the Cohen's d value of connectome hub versus the Cohen's d value of cortical thickness atrophy across 68 cortical areas for eight disorders. Positive Cohen's d value indicates thinning of cortical thickness in patients. Each dot represents one cortical area. Significance of Pearson's correlation coefficients was determined by 1,000 permutation tests and was labeled with uncorrected p value.*

2.7. “Information flow along the primary visual, visual association, and higher-level sensorimotor cortices is undertaken by the four occipital hubs (Cluster II) left VMV1, right V4, and bilateral V3A that are all densely connected with the VIS and portions of the SMN, DAN, and VAN.” This sentence needs the support of references.

Response: We have added a reference to make it clear.

Main Manuscript, Discussion, page 9, lines 321-327:

[...] Information flow along the primary visual, visual association, and higher-level sensorimotor cortices is undertaken by the four occipital hubs (Cluster II) left VMV1, right V4, and bilateral V3A that are all densely connected with the VIS and portions of the SMN, DAN, and VAN. This is supported by the report of their dense connections with both the visual system and SMN region the frontal eye field, DAN region the superior parietal cortex, and VAN region the parietal operculum and anterior insula⁵⁵ and also aligns with the role of their homologous regions in the non-human primate cerebral cortex⁵⁴.

2.8. There needs to be a discussion of the relationship between FCS maps and ENIGMA-derived patterns of brain atrophy across neurological, psychiatric, and neurodevelopmental disorders and what will be useful in clinical practice and informing the design of future studies.

Response: We thank the reviewer for this valuable suggestion. We have added an association analysis between the Cohen's d value of connectome hub and the Cohen's d value of cortical thickness atrophy in eight neuropsychiatric disorders using summarized dataset provided by the ENIGMA Toolbox²⁸.

Main Manuscript, Introduction, page 3, lines 73-77:

[...] To uncover the genetic signatures underlying these connectome hubs, we conducted machine learning approaches to distinguish connectome hubs from non-hubs using transcriptomic data from the Allen Human Brain Atlas (AHBA), explored their developmental evolutions using the BrainSpan Atlas, and assessed their neural relevance by contextualizing them relative to established neuroimaging patterns.

Main Manuscript, Results, page 8, lines 270-284:

In addition, we also noted that the above 150 key genes are enriched for several psychiatric disorders ($FE = 3.5$, uncorrected $p = 5.5 \times 10^{-4}$, Table S5). This finding is in accordance with prior observations of hub regions being preferentially targeted by neuropsychiatric disorders⁵⁻⁸. This implies that connectome hubs may have different susceptibility to neuropsychiatric disorders in contrast to non-hubs. We validated it by performing an association analysis between the effect size of connectome hub and the effect size of cortical thickness atrophy in neuropsychiatric disorders⁵⁰. We observed that the Cohen's d of connectome hub is negatively correlated with the Cohen's d of cortical thickness atrophy in 22q deletion syndrome (Pearson's $r = -0.292$, uncorrected $p = 0.009$) and autism spectrum disorder (Pearson's $r = -0.333$, uncorrected $p = 0.019$) but positively correlated with the Cohen's d of cortical thickness atrophy in bipolar disorder (Pearson's $r = 0.418$, uncorrected $p = 0.003$) and schizophrenia (Pearson's $r = 0.247$, uncorrected $p = 0.040$) (Figure 6D). This suggests that connectome hubs have a trend of higher susceptibility to cortical thickness atrophy in bipolar disorder and schizophrenia but lower susceptibility to cortical thickness atrophy in 22q deletion syndrome and autism spectrum disorder than non-hubs.

Main Manuscript, Discussion, page 11, lines 379-382:

[...] This is in line with ~~our observation of~~ the result of several psychiatric disorders being the most significant disease associated with the top 150 key genes and our observation of differences in susceptibility to cortical thickness atrophy in neuropsychiatric disorders between hub and non-hub regions. [...]

Main Manuscript, Methods, page 15, lines 541-550:

The Cohen's d value of cortical thickness atrophy in neuropsychiatric disorders was assigned to 68 cortical areas in a standard brain surface space⁵⁰. We first resampled the unthresholded Cohen's d map of connectome hub in Figure 2B from the MNI volume space to the standard brain surface space provided by the dataset⁵⁰ and computed the Cohen's d value for each of the 68 cortical areas by averaging Cohen's d value across

vertices within each cortical area. Then, we computed Pearson's correlation coefficient between the Cohen's d value of connectome hub and the Cohen's d value of cortical thickness atrophy across 68 cortical areas for each of the eight disorders (Figure 6D). To reduce the potential effects of development on our results, we used cortical thickness atrophy data from adults for the attention deficit hyperactivity disorder, bipolar disorder, major depressive disorder, and obsessive-compulsive disorder.

Main Manuscript, Methods, page 15, lines 564-570:

[...] To determine the statistical significance of Pearson's correlation coefficients in Figure 6D, we constructed 1,000 surrogate maps of the unthresholded Cohen's d map in Figure 2B with the spatial autocorrelations being corrected using a generative model⁷⁵ and repeated calculating Pearson's correlation coefficients using these surrogate Cohen's d maps to construct a null distribution. Then, p values of these Pearson's correlation coefficients were determined by comparing the observed values with their corresponding null distributions.

Main Manuscript, Figures, pages 32-33, lines 906-918:

Figure 6. Neural contextualization of connectome hubs' transcriptomic pattern. A-C

*Differences between hub (red) and non-hub (blue) regions in density of neurotransmitter receptor and transporter (A, hub voxels n=15,461, non-hub voxels n=32,158), fiber number for different fiber length bins (B, hub vertices n=25,944, non-hub vertices n=33,195), and metabolic rate for oxygen, aerobic glycolysis, and blood supply (C, hub regions n=29, non-hub regions n=60). For each violin plot, dashed gray lines depict the 25th and 75th percentiles, solid gray line depicts median value. Significance of one-sided Wilcoxon rank-sum tests was determined by 1,000 permutation tests and was labeled with Bonferroni-corrected p value. *p<0.05, **p<0.01, ***p<0.001. a.u., arbitrary unit. D Regression plot of the Cohen's d value of connectome hub versus the Cohen's d value of cortical thickness atrophy across 68 cortical areas for eight disorders. Positive Cohen's d value indicates thinning of cortical thickness in patients. Each dot represents one cortical area. Significance of Pearson's correlation coefficients was determined by 1,000 permutation tests and was labeled with uncorrected p value.*

References used in this response letter:

1. de Reus MA, van den Heuvel MP. The parcellation-based connectome: Limitations and extensions. *NeuroImage* **80**, 397-404 (2013).
2. Bassett DS, Brown JA, Deshpande V, Carlson JM, Grafton ST. Conserved and variable architecture of human white matter connectivity. *NeuroImage* **54**, 1262-1279 (2011).
3. Buckner RL, *et al.* Cortical Hubs Revealed by Intrinsic Functional Connectivity: Mapping, Assessment of Stability, and Relation to Alzheimer's Disease. *The Journal of Neuroscience* **29**, 1860 (2009).
4. Cole MW, Pathak S, Schneider W. Identifying the brain's most globally connected regions. *NeuroImage* **49**, 3132-3148 (2010).
5. Tomasi D, Volkow ND. Functional connectivity density mapping. *Proc Natl Acad Sci USA* **107**, 9885-9890 (2010).
6. Fransson P, Aden U, Blennow M, Lagercrantz H. The Functional Architecture of the Infant Brain as Revealed by Resting-State fMRI. *Cerebral Cortex* **21**, 145-154 (2011).
7. Tomasi D, Volkow ND. Association between Functional Connectivity Hubs and Brain Networks. *Cerebral Cortex* **21**, 2003-2013 (2011).
8. de Pasquale F, *et al.* The connectivity of functional cores reveals different degrees of segregation and integration in the brain at rest. *NeuroImage* **69**, 51-61 (2013).
9. Liao X-H, *et al.* Functional brain hubs and their test–retest reliability: A multiband resting-state functional MRI study. *NeuroImage* **83**, 969-982 (2013).
10. Dai Z, *et al.* Identifying and Mapping Connectivity Patterns of Brain Network Hubs in Alzheimer's Disease. *Cerebral Cortex* **25**, 3723-3742 (2014).
11. Fornito A, Zalesky A, Bullmore ET. Network scaling effects in graph analytic studies of human resting-state FMRI data. *Frontiers in Systems Neuroscience* **4**, 22 (2010).
12. Eickhoff SB, Yeo BTT, Genon S. Imaging-based parcellations of the human brain. *Nature Reviews Neuroscience*, (2018).

13. Power JD, Schlaggar BL, Lessov-Schlaggar CN, Petersen SE. Evidence for Hubs in Human Functional Brain Networks. *Neuron* **79**, 798-813 (2013).
14. Achard S, Salvador R, Whitcher B, Suckling J, Bullmore E. A Resilient, Low-Frequency, Small-World Human Brain Functional Network with Highly Connected Association Cortical Hubs. *The Journal of Neuroscience* **26**, 63-72 (2006).
15. Wang X, Lin Q, Xia M, He Y. Differentially categorized structural brain hubs are involved in different microstructural, functional, and cognitive characteristics and contribute to individual identification. *Human Brain Mapping* **39**, 1647-1663 (2018).
16. Burt JB, Helmer M, Shinn M, Anticevic A, Murray JD. Generative modeling of brain maps with spatial autocorrelation. *NeuroImage* **220**, 117038 (2020).
17. Fulcher BD, Arnatkeviciute A, Fornito A. Overcoming false-positive gene-category enrichment in the analysis of spatially resolved transcriptomic brain atlas data. *Nature Communications* **12**, 2669 (2021).
18. Chen T, Guestrin C. XGBoost: A Scalable Tree Boosting System. *arXiv*, (2016).
19. Bajada CJ, Schreiber J, Caspers S. Fiber length profiling: A novel approach to structural brain organization. *NeuroImage* **186**, 164-173 (2019).
20. Choi EY, Yeo BT, Buckner RL. The organization of the human striatum estimated by intrinsic functional connectivity. *Journal of Neurophysiology* **108**, 2242-2263 (2012).
21. Hwang K, Bertolero MA, Liu WB, D'Esposito M. The Human Thalamus Is an Integrative Hub for Functional Brain Networks. *The Journal of Neuroscience* **37**, 5594-5607 (2017).
22. Greene DJ, *et al.* Integrative and Network-Specific Connectivity of the Basal Ganglia and Thalamus Defined in Individuals. *Neuron* **105**, 742-758 e746 (2020).
23. Gratton C, *et al.* Functional Brain Networks Are Dominated by Stable Group and Individual Factors, Not Cognitive or Daily Variation. *Neuron* **98**, 439-452 e435 (2018).
24. Marek S, *et al.* Spatial and Temporal Organization of the Individual Human Cerebellum. *Neuron* **100**, 977-993 e977 (2018).

25. Liu J, Xia M, Wang X, Liao X, He Y. The spatial organization of the chronnectome associates with cortical hierarchy and transcriptional profiles in the human brain. *NeuroImage* **222**, 117296 (2020).
26. Arnatkeviciute A, Fulcher BD, Fornito A. A practical guide to linking brain-wide gene expression and neuroimaging data. *NeuroImage* **189**, 353-367 (2019).
27. Dukart J, *et al.* JuSpace: A tool for spatial correlation analyses of magnetic resonance imaging data with nuclear imaging derived neurotransmitter maps. *Human Brain Mapping* **42**, 555-566 (2021).
28. Lariviere S, *et al.* The ENIGMA Toolbox: multiscale neural contextualization of multisite neuroimaging datasets. *Nature Methods* **18**, 698-700 (2021).

REVIEWERS' COMMENTS:

Reviewer #1 (Remarks to the Author):

The authors have addressed the issues I raised with care and consideration. I congratulate them on their efforts and look forward to seeing this formally published!

Reviewer #2 (Remarks to the Author):

The authors have addressed all my concerns. Congratulations!